# LineFlow: A Framework to Learn Active Control of Production Lines

Kai Müller [1]   Martin Wenzel [1]   Tobias Windisch [1]

## Abstract

Many production lines require active control mechanisms, such as adaptive routing, worker reallocation, and rescheduling, to maintain optimal performance. However, designing these control systems is challenging for various reasons, and while reinforcement learning (RL) has shown promise in addressing these challenges, a standardized and general framework is still lacking. In this work, we introduce LineFlow, an extensible, open-source Python framework for simulating production lines of arbitrary complexity and training RL agents to control them. To demonstrate the capabilities and to validate the underlying theoretical assumptions of LineFlow, we formulate core subproblems of active line control in ways that facilitate mathematical analysis. For each problem, we provide optimal solutions for comparison. We benchmark state-of-the-art RL algorithms and show that the learned policies approach optimal performance in well-understood scenarios. However, for more complex, industrial-scale production lines, RL still faces significant challenges, highlighting the need for further research in areas such as reward shaping, curriculum learning, and hierarchical control.

## 1. Introduction

At its core, manufacturing is about transforming raw materials into finished goods, often on a large scale. In most production systems, the necessary process steps are carried out by work stations which each component has to traverse sequentially. Typically, the work steps necessary have to be applied in a fixed order, creating an interdependence between stations in the sense that performance issues of one station directly affect stations down- and upstream.

In theory, finding optimal layouts for production lines that take the interdependence of the individual work steps into account is a computationally hard but well understood problem called *assembly line balancing problem* (Padrón et al., 2009; Scholl & Becker, 2006). Solution strategies are powered by good stochastic models for interlinked production lines, like in (Bierbooms, 2012; Liberopoulos et al., 2006), that help computing the equilibrium state of the throughput for a given layout. However, in practice, optimal layouts and theoretic knowledge on the equilibrium of the throughput does not guarantee that the production line unfold its full theoretical potential. This may be due to situations like changing processing conditions, adverse coincidences in the stochasticity of the processes, or machine failures. These circumstances can suddenly lead to full buffers, jams, and shifting bottlenecks (Roser et al., 2003; Mahmoodi et al., 2022). Thus, reacting in a dynamic and optimal way *at runtime* is key to keep the performance of the production line at its best. At present, this is either a completely manual task accomplished by line workers on-site or by *line control* systems, that actively adjust line parameters in real-time (see Figure 1). Possible interventions are rerouting tasks to alternative stations (Huang et al., 2019), reallocating workers (Pérez-Wheelock et al., 2022; Dolgui et al., 2018), or scheduling maintenance (Geurtsen et al., 2023). Classical

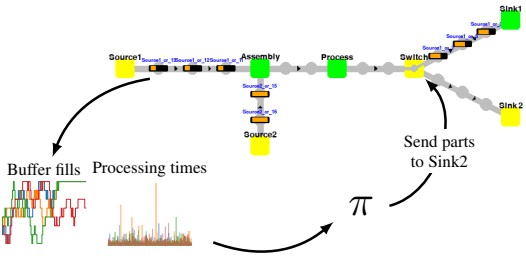

*Figure 1.* Active line control based on real-time data.

active line control systems typically combine rule-based strategies and mathematical models to optimize scheduling and resource allocation, focusing mostly on bottleneck identification, see for instance (Roser et al., 2014; 2017; Li et al., 2009; Lai et al., 2021) and references therein. Finding optimal control strategies is a complex task that not only requires a detailed understanding of the production line dynamics but also involves time-intensive trial-and-error session on the real system and simulations. While effective

---

[1]University of Applied Sciences Kempten, Kempten, Germany. Correspondence to: Tobias Windisch <tobias.windisch@hs-kempten.de>.

*Proceedings of the 42^{nd} International Conference on Machine Learning*, Vancouver, Canada. PMLR 267, 2025. Copyright 2025 by the author(s).

in predictable scenarios, classical approaches often struggle in dynamic and complex environments with high uncertainties, which has motivated the use of machine learning for tasks like predicting bottlenecks (Subramaniyan et al., 2021) and forecasting demand (Gonçalves et al., 2021).

Since active production line control is, at its core, a sequence of observing a state and taking action, it seems like a natural next step to learn line control in an end-to-end fashion through interactions with a simulation using reinforcement learning (RL). RL algorithms have demonstrated the ability to learn complex relationships between states and actions (Vinyals et al., 2019; Silver et al., 2016; Wu et al., 2023; Jumper et al., 2021). Significant research has been conducted on learning control policies for production lines (see Section 1.1); however, progress is hindered because existing studies each rely on their own ad hoc or domain-specific simulations, making benchmarking, reproducibility, and the assessment of RL progress in active line control difficult. One reason for this is the absence of a theoretical or practical framework to train and evaluate RL agents in various production settings in a standardized way.

In this work, we address exactly this research gap by introducing LineFlow[1], a free and extensible Python package specifically tailored to simulate production lines in order to train RL agents for active line control at scale (Section 2). LineFlow not only is a fine-graded and fast discrete-event simulation of production lines, it also comes with an implementation of conventional action- and state-spaces as well as a flexible score (Section 2.3) inspired by classical performance indicators that allows to train agents for various production settings (Section 3). To evaluate its theoretic underpinnings, we study well understood and realistic production scenarios by computing their mathematical optimum (Section 4) and then comparing it with the performance of control policies interacting with LineFlow (Section 5). To showcase the current limits, we also analyze a very complex production setting as mentioned above with unclear optimum and train RL agents via curriculum learning (Section 4.4). As there exist strong privacy concerns among manufacturers when it comes to releasing production line layouts, publicly available datasets holding performance information and non-trivial layouts are rare. Here, we hope that LineFlow will close this gap by allowing manufacturers to provide synthetic non-confidential digital twins preserving their key runtime challenges to the public. In Section D, we model a real-world production line based on publicly available data in LineFlow. Our research is purely application-driven by the needs of end-users in manufacturing and opens up a wide range of challenges for machine learning researchers to investigate. Since implementing active control systems is currently a highly manual task guided

by domain expertise, we aim for LineFlow to accelerate research on learning active line control in the future.

## 1.1. Related Work

There has been a lot of research where RL methods are studied to control a given manufacturing technology, like for welding (Jin et al., 2019; Masinelli et al., 2020; Zou & Lan, 2020) or molding (Szarski & Chauhan, 2023; Guo et al., 2019) just to name a few. We refer to (Nian et al., 2020) and references therein for an overview. In contrast, our work treats the industrial technology happening inside the processes as a black box and solely considers the *processing time* of the stations and their statistical interplay. This means, we neither model physical relations nor monitor physical parameters of the processes, which is a fruitful application field for RL on its own (Viharos & Jakab, 2021).

A large body of research directly related to our work has been done for production scenarios where tasks of varying length have to be scheduled on heterogeneous machines with different processing capabilities or resources. Such *scheduling problems* are tackled with RL-based approaches in many works, like in (Shi et al., 2020; Shiue et al., 2018; Kim & Lee, 1998; Tortorelli et al., 2022; Ali & Tirel, 2023; Overbeck et al., 2021). A comprehensive overview is provided in (Kuhnle et al., 2021). Other research focuses on machine interactions, as in (Wang et al., 2016), where RL is used to control a layout consisting of two stations and one buffer to maximize throughput or in (Loffredo et al., 2024; 2023), where energy costs are minimized by switching stations dynamically to standby mode. In (Geurtsen et al., 2023), the scheduling of predictive maintenance of a production line using RL has been studied.

## 2. Simulating Production Lines

Production lines can be characterized in many ways, like by the types of their processes or parts, their production volume, or how their stations are arranged (Kang et al., 2020; Liberopoulos et al., 2006). Here, we focus on production lines that produce discrete items. This section introduces general principles and mechanisms of such production systems and explains how these are reflected in LineFlow.

## 2.1. Objects of Production Lines

The main objective of production lines studied in this work is to produce discrete items called *parts*. Typically, the production of a single part requires a specific set of work steps that need to be applied in a certain order involving the production and assembly of a series of sub-*components*. Each work step is carried out in a specific *station*. One of the objectives of LineFlow is to model the statistical dynamics of a production line by considering the individual processing

---

[1] https://github.com/hs-kempten/lineflow.

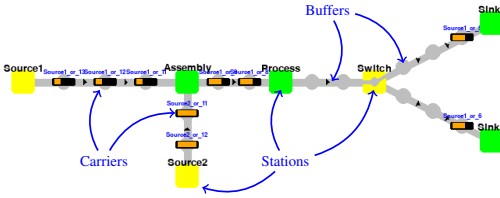

*Figure 2.* A production line visualized with LineFlow.

times of the stations and their interplay. To be more precise, the processing times are assumed to be exponentially distributed like in (Bierbooms, 2012) as $\mathcal{T} = T + \mathrm{Exp}_S$, where $T \geq 0$ is the minimal processing time possible and where $\mathrm{Exp}_S$ denotes the exponential distribution[2] with mean $S$. We distinguish different types of stations, among them are *sources* that set up components, *processes* that apply a piece of work on a single component, *assemblies* that join two or more components into one, and *sinks* that remove a fully build part from the line (see Figure 2). Components are transported by *carriers* from station to station via *buffers*, which work on a first-in-first-out principle. Buffers can only hold a predefined number of carriers, called their *capacity*. Time-intense work steps may be distributed over identical parallel stations, while *switches* handle the routing of carriers. The arrangement of stations and buffers is typically called the *layout* of the production line.

## 2.2. Active Line Control

The overall performance of a production line depends sensitively on the individual performances of its stations as performance issues of any station are propagated down- and upstream by their linking buffers. For instance, consider the production line shown in Figure 2: If the processing time of *Process* is higher than the processing time of *Assembly*, the buffer in between fills up and once its maximal capacity is reached, the previous station is blocked as it cannot push finished components to the buffer. Such a jam can be propagated backwards through the line and may even block the source. Similarly, subsequent stations like the two sinks with faster processing times are not served with components and have to wait till new components arrive. In the equilibrium, the performance of the full system depends essentially on the bottleneck station, which is in simple cases just the station having the highest processing time. Buffers constitute an important parameter of production lines as they can dampen fluctuations in processing times at the stations. However, buffers and their capacity are fixed parameters of the layout, whereas stochasticity in the processing times of the processes can result in a shifting runtime bottleneck (Roser et al., 2003). Thus, the bottle-

neck can vary, whereas the buffer capacities cannot. We refer to (Bierbooms, 2012) for a detailed analysis of the interplay of stations and buffers. Bottlenecks in production lines do not only reduce throughput by slowing down the overall part flow, but they can also lead to quality issues and even scrap. One prominent example arises in processes with strict timing constraints between consecutive steps. In adhesive bonding, for instance, components must be joined within a certain time window after the adhesive is applied. If a downstream bottleneck causes delays and this threshold is exceeded, the bond can no longer be guaranteed to meet quality standards, and the affected part must be discarded (Chen et al., 2005). Such timing-sensitive constraints make bottleneck prevention critical not only for efficiency, but also for product quality and material waste reduction.

To circumvent or mitigate such harmful runtime effects, we essentially study three corrective actions and their interplay: Changing worker assignments to speed up bottleneck stations, changing the distribution of components, and including waiting times to prevent scrap. On the one hand, increasing the waiting time is an important counter activity to prevent scrap, but there is a thin line between preventing scrap and slowing down the bottleneck and, in turn, the whole line (see Section 4.1). Switches, on the other hand, allow to change the routing and distribution of components to keep the bottleneck loaded constantly. For instance, if the processing time of *Sink1* starts to increase, the switch could change the distribution ratio between the sinks (see Section 4.2). Finally, reassigning workers from one station to another directly impacts their processing time and can help to handle peak loads but can create a bottleneck elsewhere (see Section 4.3). A more involved scenario where all these actions have to be combined effectively is studied in Section 4.4. Even if a bottleneck is correctly detected, choosing an optimal action sequence to mitigate is challenging since actions take time to show results. For example, reassigning a worker requires task completion and relocation first. While this reduces the bottleneck's mean processing time, high variance makes changes hard to detect. In general, even with the right actions, high data variability delays recognizing improvements. Compounding this issue, production lines often operate near equilibrium states, where minor disruptions can cascade into buffer overflows, bottlenecks, or system-wide jams. Moreover, the combination of different actions may have unintended negative consequences on the performance. For instance, if *Sink1* is the current bottleneck, then assigning more workers to *Sink1* and sending more components to *Sink2* additionally, may result in a worse overall performance. To apply optimal actions, line control systems rely on real-time data from various production line objects, which is often noisy, incomplete, and lagged (see Section 3.2 and Section A.7).

---

[2]In the literature typically denoted as $\mathrm{Exp}_{\frac{1}{\lambda}}$.

## 2.3. Performance Measurements

There are several ways to measure the performance of production lines. A commonly used metric is the *Overall Equipment Effectiveness*, short OEE, which quantifies the effectiveness of equipment and machinery used (Nakajima, 1988). Assume we have given a control policy $\pi$ for a production line decomposed into stations $P_1, \ldots, P_k$. Denote by $n_{\text{ok}}^{\pi}(t, i)$ and $n_{\text{nok}}^{\pi}(t, i)$ the cumulated number of OK and not OK (NOK) parts produced at station $P_i$ at time $t$. Then, following (Nakajima, 1988), the OEE for $P_i$ is essentially defined as $\text{OEE}_i(t) = \frac{T_i}{t} \cdot n_{\text{ok}}^{\pi}(t, i)$ where $T_i$ is the minimum of the processing time of $P_i$. By construction, we have $n_{\text{ok}}^{\pi}(t, i) \cdot T_i \leq t$ for all $t$ and thus the maximal possible OEE is 1. Analogously, the OEE of a production line can be defined. The OEE definition is mainly focusing on productivity and quality, but a more economic view requires to also take costs of the individual stations and components into account, for instance as some stations require more energy for their processing, like in (Loffredo et al., 2024), or because only some components are scraped, not the full part. Thus, we assume that applying work at station $P_i$ comes with a cost $c_i$, which may be material costs for joining a component or costs for energy as in (Loffredo et al., 2024). The value of the final part is considered to be the sum of all costs $c = \sum_{i=1}^{k} c_i$. At simulation time $t$, the production has produced an aggregated value of

$$C_\pi(t) = \frac{T_C}{t} \left( c \cdot n_{\text{ok}}^{\pi}(t) - \sum_{i=1}^{k} c_i \cdot n_{\text{nok}}^{\pi}(t, i) \right) \quad (1)$$

where $n_{\text{ok}}^{\pi}(t) \in \mathbb{N}$ denotes the number of parts produced at all sinks combined and $T_C$ denotes the minimal possible time the line needs to produce a single part. One can think of Equation 1 as the value the production line generates: If $C_\pi(t) > 0$, the line generates revenue, if $C_\pi(t) < 0$, it generates costs. If no scrap can occur, maximizing $C_\pi(t)$ is equivalent to maximize the number of produced parts.

Generally, one wants to find a control policy $\pi$ that maximizes $C_\pi(t)$ and *simultaneously* minimizes the time $t$ to reach the maximum. It then depends on the exact usecase to weight a smaller $t$ needed with a higher $C(t)$ reached by. To circumvent this distinction in our work and to create a general benchmarking environment, one either has to fix the available time $T_{\text{sim}}$ and to maximize the output $C_\pi(T_{\text{sim}})$, or to fix the desired output $\tilde{C}$ and to minimize the time needed by a policy $\pi$ to reach it. In our benchmark studies, we decided to fix $T_{\text{sim}}$ as this typically originates from high-volume production where the output needs to be maximized in a given work shift of length $T_{\text{sim}}$.

## 3. RL for Active Line Control

In this section, we describe how RL can be used to learn policies for active line control. Specifically, we consider active line control as an *episodic* and *partially-observed* Markov decision process. Let $\mathcal{S}$ be a set of states and $\mathcal{A}$ a set of actions. Transition probabilities $P$ dictate how likely transition from a state $s \in \mathcal{S}$ to a state $s' \in \mathcal{S}$ is when choosing action $a \in \mathcal{A}$ at $s$, namely $P(s'|s, a)$. When selecting $a$ at $s$, a reward $R(s, a)$ is observed. Additionally given a desired length $T \in \mathbb{N}$, this gives a *Markov decision process* $(\mathcal{S}, \mathcal{A}, P, R, T)$ where the goal is to find a *policy* $\pi : \mathcal{S} \to \mathcal{A}$ to maximize $\mathbb{E}_{P,\pi} \left[ \sum_{t=0}^{T} R(s_t, \pi(s_{t-1})) \right]$ where $(s_t)_{t \in \mathbb{N}}$ is chosen under the interplay of $P$ and $\pi$. If the states $s \in \mathcal{S}$ cannot be observed directly by the policy but only features $f \in \mathcal{F}$ instead controlled by another conditional probability density function $O$, i.e., the likelihood of observing $f$ at state $s$ is $O(f|s)$, then the tuple $(\mathcal{S}, \mathcal{A}, P, R, \mathcal{F}, O)$ defines a *partially-observed* Markov decision process where a policy $\pi : \mathcal{F} \to \mathcal{A}$ needs to be found maximizing $\mathbb{E}_{P,O,\pi} \left[ \sum_{t=0}^{T} R(s_t, \pi(f_{t-1})) \right]$. Note that the definitions given here are streamlined versions of the common way of defining Markov decision processes, where the process typically ends in a set of defined terminal states and additionally, a *discount factor* $\gamma \in [0, 1]$ is given to trade-off rewards in early and late states.

While the formulation above assumes a discrete-time decision process, we emphasize that the underlying dynamics of the production line still can be *continuous time* and only the agent interaction—i.e., observation and action execution—is restricted to fixed intervals. This modeling choice reflects real-world production control, where decisions like worker reallocation or routing changes are typically made at regular intervals, such as every few seconds or at shift-level granularity. This approach offers a practical compromise between physical realism and learning efficiency, and is consistent with prior work in industrial reinforcement learning. Section 3.3 provides details how this is realized in LineFlow.

### 3.1. Episodes and Rewards

As described in Section 2.3, we consider constant operation times $T_{\text{sim}}$ of the manufacturing setting to be optimized. In principle, a control policy can interact with a production line at any time. We, however, assume that the policy can only interact with the production line at fixed and equidistant time points each $T_{\text{step}}$ apart, thus we get $T = \frac{T_{\text{sim}}}{T_{\text{step}}}$ many interactions in an episode. This approach introduces the possibility that the model may miss certain state changes occurring between observation intervals, such as a very brief buffer overflow. However, from a RL point of view, this has multiple advantages: First, the trajectories all become of fixed length and second, the policy does not need the time scale.

Allowing arbitrary interactions of a policy, particularly at the early stage of the training, can lead to many thousands of interactions within only a few time units. Although in LineFlow, $T_{\text{step}}$ and $T_{\text{sim}}$ are considered to be constants coming from the situation to be optimized, both can be varied sequentially in a curriculum learning fashion (Narvekar et al., 2020). The goal of an episode is to find a policy $\pi$ that maximizes $C_\pi(T_{\text{sim}})$. To allow temporal difference learning (see (Sutton & Barto, 2020, Chapter 6)), we decompose $C_\pi$ into discrete quantities for $t \in \{0, \ldots, T_{\text{sim}}\}$ by defining

$$R(s_t, \pi(s_{t-1})) = C_\pi(T_{\text{step}} \cdot (t+1)) - C_\pi(T_{\text{step}} \cdot t).$$

That way, we get $\sum_{i=0}^{T} R(t) = C_\pi(T_{\text{sim}})$ as desired.

### 3.2. States, Observations, and Actions

Due to the stochastic nature of production processes and sensor inaccuracies, the true system state of a production line is not directly observable. Instead, it must be inferred from incomplete, noisy, and delayed observations. Key features relevant for performance estimation include buffer fill levels, station processing times, production rates, station modes, and routing information from switches (Roser et al., 2003; 2014). Whenever available, features in LineFlow are constrained by known upper and lower bounds. For example, fill levels are normalized to $[0, 1]$, while production rates or processing times are positive but unbounded.

The action space in LineFlow primarily consists of discrete control decisions, such as assigning workers or routing components across stations. Additionally, stations can be switched on or off, similar to (Loffredo et al., 2023). Workers are drawn from predefined pools associated with specific station groups. Each worker is modeled as an individual and independent dimension in the action space, where the possible values correspond to the stations the worker can be assigned to. While this introduces symmetries in the action space when workers are indistinguishable, it enables fine-grained modeling of worker attributes—such as skill levels or expertise—when they are not. Furthermore, representing workers as explicit objects within the production line ensures that side constraints, such as a fixed number of available workers, are respected at all times. Switches typically feature two discrete actions: one controlling incoming buffers and another for outgoing buffers. Beyond the core actions, LineFlow also supports modifications to the physical production environment, such as actively adding or removing carriers from the line. While not included in our case studies, these actions provide additional flexibility for modeling dynamic settings. A list of all observable and actionable dimensions is in Section A.7.

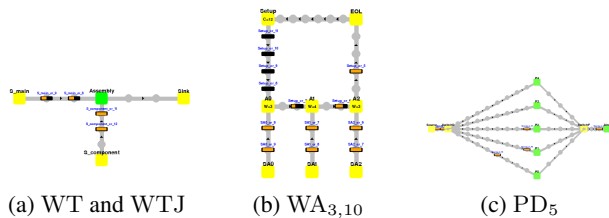

(a) WT and WTJ    (b) WA$_{3,10}$    (c) PD$_5$

*Figure 3.* Representatives of the three atomic production line challenges analyzed in this case study.

### 3.3. Implementation Details

LineFlow is designed to address key challenges in optimizing active line controls while meeting the needs of RL researchers. The discrete event simulation, handling station interactions and stochasticity, is built on SimPy (SimPy, 2025). Its object-oriented structure enables easy customization of stations and rapid setup of complex production scenarios. A visualization module based on pygame provides insights into layout dynamics and agent interactions. Fully implemented in Python, LineFlow integrates seamlessly with data science tools like pandas (Wes McKinney, 2010) and numpy (Harris et al., 2020). RL interaction follows the gymnasium API (Towers et al., 2024), allowing training with stable-baselines3 (Raffin et al., 2021) or skrl (Serrano-Muñoz et al., 2023). Environments support vectorization and parallelization to accelerate training. Further details are provided in Section A.1, with a complete example in Section A, illustrating the layout in Figure 2. Although agent interaction follows the discrete-time process described in Section 3, the underlying production dynamics in LineFlow is simulated in continuous time using a discrete-event engine. The interaction frequency of the agent can be adjusted via a parameter, allowing the discrete-time control to approximate continuous-time behavior with high fidelity.

## 4. Case Studies

In this section, we introduce and study realistic production scenarios where active control is required. The scenarios have been selected for two reasons: First, their theoretical optimum can be computed statistically allowing us to quantify whether RL algorithms learn optimal policies. Second, they appear as subproblems in many production scenarios.

### 4.1. Optimal Waiting Time WT and WTJ

In this scenario, the optimal waiting time between parts produced by a source station $S_C$ has to be found (see also Figure 3a). The source $S_C$ serves together with another source $S_M$ an assembly station $A$, which joins the components from both sources to produce the final product and sends it to sink $S$. Components from source $S_C$ have a fixed

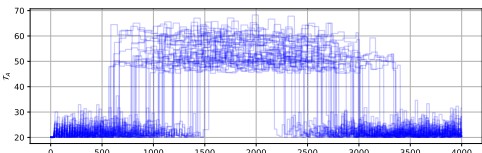

*Figure 4.* The jumps in the processing time of the assembly $A$ in WTJ for different simulations of length 4000.

expiration time $T_{\mathrm{AC}}$ called *assembly condition*: If the time from their setup at $S_C$ to the time the process starts at $A$ is larger than $T_{\mathrm{AC}}$, then $A$ has to dispose this component and has to wait for the next one. The removal of a scrap part not only generates costs, it also blocks the assembly for a certain amount of time as the defective component has to be removed first. Such a situation is typical in many joining processes, like gluing applications, where adhesives can dry out if not processed in time (Chen et al., 2005). If the waiting time is chosen too small, $S_C$ produces to many components which cannot be handled by $A$ in time and thus drives scrap costs. On the other hand, if the waiting time is chosen to high, $A$ has to wait for components which delays the production of final parts. The goal is to balance the waiting time of $S_C$ to maintain a continuous component supply at $A$. Assuming costs $c_M$ and $c_S$ for components produced at $S_M$ and $S_C$ respectively and that only components of $S_C$ can potentially be scraped, the performance of the line as described in Section 2.3 within a given time frame $t$ is

$$C_\pi(t) = \frac{T_C}{T_{\mathrm{sim}}} \left( (c_M + c_S) \cdot n_{\mathrm{ok}}^\pi(t) - c_S \cdot n_{\mathrm{nok}}^\pi(t, A) \right).$$

We generally assume that the processing times without potential waiting times are such that $A$ is the bottleneck of the line. Thus, the maximum number of parts produced depends on the time $A$ needs to get one carriers and one component, to assemble them, and to push the final product to the buffer. The optimal waiting time essentially fills the between the times $A$ and $S_C$ need to handle and process their parts. We give an explicit equation for the optimal waiting time and the maximum number of parts in Section B.1.

To force a dynamic adaption we now introduce the related scenario *Waiting time jump* (WTJ). Here, we uniformly sample a length $T_{\mathrm{jump}}$ and a trigger time $T_{\mathrm{trigger}}$ such that $[T_{\mathrm{trigger}}, T_{\mathrm{trigger}} + T_{\mathrm{jump}}] \subset [0, T_{\mathrm{sim}}]$. Then, the processing time of $A$ at simulation time $t$ from

$$\begin{cases} T + \mathrm{Exp}_S, & \text{if } t \notin [T_{\mathrm{trigger}}, T_{\mathrm{trigger}} + T_{\mathrm{jump}}] \\ f \cdot T + \mathrm{Exp}_S, & \text{if } t \in [T_{\mathrm{trigger}}, T_{\mathrm{trigger}} + T_{\mathrm{jump}}] \end{cases}.$$

As $f$ and $T_{\mathrm{jump}}$ can be different in each episode, the maximal possible reward can vary, too. To ease the comparison of agents over multiple runs, we construct $f$ for given $T_{\mathrm{jump}}$

in a way that the maximal possible reward remains a constant (see Section B.1 for details). More precisely, we fix a constant $0.5 < R < 1.0$ and construct $f$ such that expected maximal parts produced is $R \cdot N$, where $N$ is the expected number of parts of WT without jump. A visualization of the processing time of $A$ for multiple simulations is in Figure 4. Although, by design, the expected maximal number of parts produced by the line is $R \cdot N$, this value cannot be reached by a control agent. The reason is that at time $T_{\mathrm{trigger}}$ if the processing time of the assembly jumps to a higher level, the new processing time of $A$ can first be observed once the first part has been produced. Thus, from $T_{\mathrm{trigger}}$ to $T_{\mathrm{trigger}} + f \cdot T + \mathrm{Exp}_S$, the source $S_C$ runs with a too low waiting time and possibly sends components to $A$ which are going to expire. The same happens once the processing time of $A$ jumps to the lower level. Here, the source may run at a higher waiting time causing $A$ to wait. To quantify the optimum reachable by agents learning from observations, we evaluate the performance of an agent learning the means of the processing times online (see Section B.1).

### 4.2. Optimal Part Distributions $\mathrm{PD}_k$

Our next case study is a subproblem of scheduling problems in manufacturing and is already extensively researched in literature (see Section 1.1). Here, components have to be distributed onto $k$ many parallel processes $P_1, \ldots, P_k$ with processing time distributions $\mathcal{T}_1, \ldots, \mathcal{T}_k$ by a single switch in an optimal way. Moreover, another switch needs to fetch the outputs of the processes and sends them to a sink (see Figure 3c). We assume that $\mathcal{T}_i = T_i + \mathrm{Exp}_{S_i \cdot T_i}$ where $T_i$ and $S_i$ are constants for all $i \in [k]$. To compute the optimum, we assume for simplicity that get and put times from the connecting buffers are already contained in $\mathcal{T}_i$. Within a given time frame $T_{\mathrm{sim}}$, the expected maximal number of parts $\mathcal{N}_i$ produced by process $i$ is obtained when $P_i$ is served with parts constantly without time gaps: $\mathbb{E}[\mathcal{N}_i] = \frac{T_{\mathrm{sim}}}{\mathbb{E}[\mathcal{T}_i]} = \frac{T_{\mathrm{sim}}}{(1 + S_i) \cdot T_i}$. When we assume that the processing times of the source and the sink are negligible compared to the processing times of the processes $P_i$. The maximal number of parts $\mathcal{N}$ produced by the production line within $T_{\mathrm{sim}}$ is the sum of the maximal parts produced by all processes, i.e., $\mathcal{N} = \sum_{i=1}^{k} \mathcal{N}_i$ as all processes can work in parallel. This allows us to compute the maximum number of parts the layout can produce theoretically, see Section B.2. Clearly, the greedy policy deployed at both switches which pushes components on the buffer with lowest fill and fetches components from the buffer with highest fill loads the processes in an optimal way (see also Section B.2).

### 4.3. Optimal Worker Assignments $\mathrm{WA}_{k,N}$

In many production scenarios, processes involve manual work which can be completed faster by multiple workers

collaborating. In this case study, the goal is to find the right distribution of a limited number of workers over multiple stations each having a varying processing time. We assume that the required processing time of a station, where $n$ workers are assigned to, is sampled from the distribution $\mathcal{T}_{T,S,n} := T \cdot p_c(n) + \mathrm{Exp}_{S \cdot T}$ where $T, S \in \mathbb{R}_{>0}$ are constants and where $p_c(n) = \exp(-c \cdot n)$ is a *performance coefficient* (see also Section B.3). In our benchmarks, we set $c = 0.3$, which means that an additional worker reduces the processing time by approximately $74\%$. Now, consider a layout of $k$ sequential stations $A_1, \ldots, A_k$ with constants $T_1, \ldots, T_k$ and $S_1, \ldots, S_k$. Moreover, assume the line has a pool of $N$ workers to distribute. As the stations are sequentially arranged and connected via buffers (see Figure 3b), the slowest process determines the speed of the overall production line. This means, a partition of $N$ into $k$ integer summands, i.e., $n_1 + n_2 + \ldots + n_k = N$ with $n_i \in \mathbb{N}_0$ needs to be found such that the maximum of all processing times is minimal. Clearly, the assignment process is bound to certain constraints, like delayed start of processes due to the waiting for assigned workers because of traversal times. We refer to Section 3.2 for more details about how the worker assignment is modeled in LineFlow. Let $\mathcal{N}_{N,k} \subseteq [N]^k$ be the set of partitions of $N$ into many $k$ non-negative integer numbers. More formally, the following max-min integer optimization problem needs to be solved:

$$T_C^* = \min_{(n_1,\ldots,n_k) \in \mathcal{N}_{N,k}} \max_{i \in [k]} \mathbb{E}[\mathcal{T}_{T_i,S_i,n_i}] \qquad (2)$$

where in our case, the expected value of the processing time is $\mathbb{E}[\mathcal{T}_{T_i,S_i,n_i}] = T_i \cdot p_c(n_i) + S_i \cdot T_i$. The maximum number of parts that can be produced is then essentially given by $\frac{T_{\mathrm{sim}} - T_{\mathrm{off}}}{T_C^*}$ after removing some initial offsets $T_{\mathrm{off}}$ due to the ramp-up of the production line. Mathematically, solving Equation (2) is computationally hard not alone because it is a min-max problem but also because it involves integer partitions. Learning an optimal distribution in an interactive environment is even more challenging, as first good estimates for $T_i$ and $S_i$ need to be learned as well as for $p_c$. In Section B.3, we formulate Equation 2 as mixed-integer optimization problem and compare the empirical results obtained with LineFlow with the theoretical optimum.

### 4.4. Complex Line CL

Finally, we combined all studied problems into a problem we call *complex line*, short CL (see Figure 5). Here the agent has to distribute workers and components over $k$ sequential assembly stations such that an assembly condition $T_{\mathrm{AC}}$ as in WT for the components is kept. Interestingly, simply combining the individual and optimal solutions of the individual subproblems does not lead to an optimal solution for CL. A straightforward distribution approach with a fixed distribution of workers, parts and a fixed waiting time at the source can cause excessive scrap or a slow buffer-

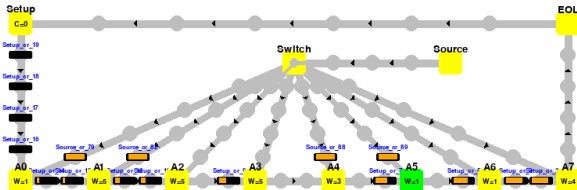

*Figure 5.* Scenario CL with 8 assemblies.

filling process between the source, switch, and assembly stations. In Section B.4, we present a profitable heuristic found by extensive testing meaning the number of produced parts outweighs the cost of scrap.

## 5. Benchmarks

### 5.1. Layout of Experiments

First, we want to emphasize that our benchmark is intended to serve as a proof of principle how evaluation can be done with LineFlow and to show that optimal control can be learned. For that, we only select a handful of algorithms and hyperparameters to consider, although many more would be feasible as well. Concretely, we use the following algorithms in our experiments: PPO (Schulman et al., 2017) and its recurrent version (Pleines et al., 2022), A2C (Mnih et al., 2016), and TRPO (Schulman et al., 2015), all of which can deal with multi-dimensional observations of mixed types and multi-dimensional categorical action spaces. We used the implementations provided by the stable-baselines (Raffin et al., 2021) package and evaluated multiple hyperparameters with three different random seeds for each case study. More details on the hyperparameters used can be found in Section C. The performance of the agents was measured in online evaluations, reporting the mean reward of 5 episodes on a separate evaluation environment 5 times (see also Figure 6 for the evolution of the reward over global steps). When not stated differently, the deterministic version of a trained policy is evaluated, that is, the action with the highest probability is selected. In all experiments, the agents are trained and evaluated on five environments stacked into a vectorized environment.

### 5.2. Results for WT, WTJ, WA, and PD

All benchmarks we set $T_{\mathrm{sim}} = 4000$. For WT and WTJ, we set $\mathcal{T}_A = 20 + \mathrm{Exp}_2$, $\mathcal{T}_{SC} = 5 + \mathrm{Exp}_{0.5}$, and use a constant get time $\mathcal{T}_g = 1$. The trigger time $T_{\mathrm{trigger}}$ is uniformly sampled from the interval $[500, 1500]$ whereas $T_{\mathrm{jump}}$ is uniformly sampled from $[1600, 2000]$. The assembly condition is set to $T_{\mathrm{AC}} = 35$. In $\mathrm{PD}_k$, the time distributions are $\mathcal{T}_i = 10 \cdot (i + 1) + \mathrm{Exp}_{0.1}$ for $i \in [k]$. In $\mathrm{WA}_{k,N}$, we set $N = 3 \cdot k$ and use $c = 0.3$ for the performance coefficient. Moreover, we set $T_i = (16 + i \cdot 4)$.

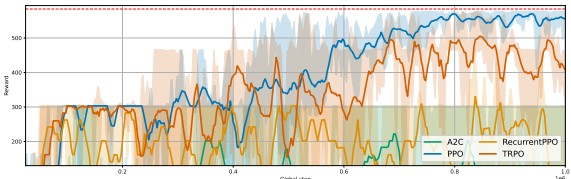

*Figure 6.* Evaluation reward over global steps for PD$_3$.

|  | **PPO** | **TRPO** | **rPPO** | **A2C** | **Optimal** |
|---|---|---|---|---|---|
| WT | $155.0 \pm 3.0$ (158.4) | $157.1 \pm 1.9$ (158.6) | $157.7 \pm 1.3$ (**159.0**) | $154.4 \pm 5.9$ (158.6) | $156.2 \pm 1.5$ |
| WTJ | $93.7 \pm 6.9$ (101.4) | $93.9 \pm 7.6$ (102.4) | $105.9 \pm 11.6$ (**113.0**) | $100.5 \pm 11.1$ (110.2) | $114.8 \pm 0.9$ |
| WA$_{3,9}$ | $274.7 \pm 13.3$ (288.6) | $244.6 \pm 18.0$ (262.8) | $273.7 \pm 21.9$ (288.4) | $278.9 \pm 17.4$ (**289.4**) | $287.1 \pm 2.7$ |
| WA$_{4,12}$ | $243.1 \pm 9.0$ (252.0) | $194.6 \pm 7.7$ (202.6) | $231.9 \pm 21.3$ (246.4) | $244.7 \pm 15.4$ (**255.0**) | $252.8 \pm 2.1$ |
| WA$_{5,15}$ | $210.5 \pm 22.0$ (**235.0**) | $151.2 \pm 6.9$ (155.6) | $211.9 \pm 27.8$ (230.0) | $207.7 \pm 38.4$ (234.2) | $236.3 \pm 2.1$ |
| PD$_3$ | $578.7 \pm 0.6$ (**579.2**) | $539.1 \pm 61.3$ (577.8) | $359.6 \pm 95.1$ (469.4) | $304.3 \pm 0.5$ (304.8) | $582.7 \pm 2.6$ |
| PD$_4$ | $651.1 \pm 9.3$ (**661.2**) | $582.1 \pm 101.5$ (657.2) | $585.8 \pm 31.7$ (620.2) | $305.1 \pm 0.1$ (305.2) | $670.7 \pm 2.2$ |
| PD$_5$ | $651.3 \pm 107.0$ (**715.6**) | $664 \pm 77.0$ (712.6) | $305.1 \pm 0.2$ (305.4) | $358.3 \pm 92.5$ (465.2) | $738.3 \pm 3.5$ |

*Table 1.* Overview of rewards obtained by the best hyperparameter combination. Number in brackets denotes maximal reward obtained (see Section B for details). Note that optimal policies are shaped by stochasticity as well and trained RL agents may outperform the expected mean performance in individual episodes.

The performance of the best hyperparameter combination for each algorithm, averaged over multiple seeds, is shown in Table 1. More details of the rewards obtained during online evaluation of all hyperparameter sets is in Section C. Scenario WT is feasible for all algorithms, but only recurrent PPO approaches the optimal value for WTJ. The actor-critic method A2C outperforms policy-based methods in WA by achieving a better performance with fewer steps. However, in part distribution, actor-critic methods fail to learn good controls while policy-based approaches succeed.

**5.3. Results for** CL

Informed decisions for CL require to keep past actions in memory. For instance, sending a component to station $i$ at a give step requires to send a part to station $i + 1$ in the one of the subsequent step in order to keep the part flow on the main track. We found that, no matter which hyper-

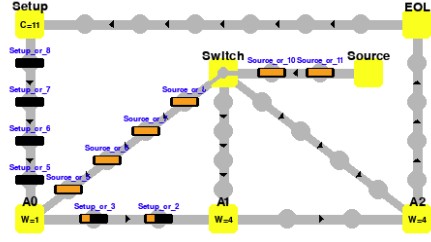

*Figure 7.* An agent trained with the PPO algorithm causing a blocking state of the line: All components send to $A_0$ causing $A_1$ to wait. As $A_1$ is blocked, the buffer between $A_0$ and $A_1$ gets full, which in turn blocks $A_0$ and consequently the full line.

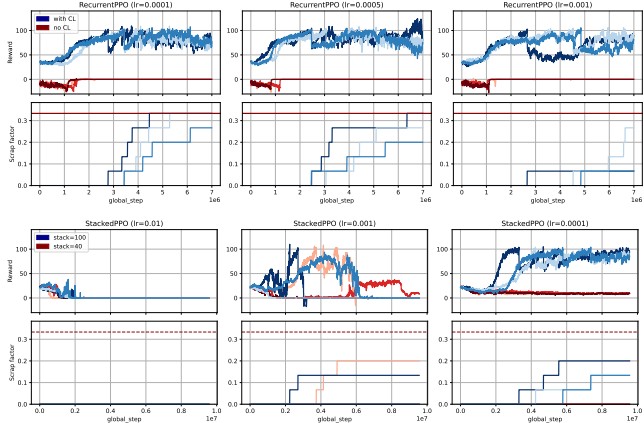

*Figure 8.* Upper figure shows a comparison of recurrent PPO trained with (blue) and without (red) curriculum using three different seeds. Lower figure shows a comparison of the stacked PPO trained using curriculum with a stack of 40 (red) and 100 (blue).

parameters have been used, hardly an algorithm reached a strictly positive reward. Since, at the start of the training the number of scrap parts heavily outweighs the number of produced parts the reward starts significantly negative. Interestingly the agents all converged to a reward of zero. An investigation revealed that agents learn to dead-lock the line, leading to no produced parts but also, no scrap parts (see Figure 7). From there, the same train data is generated during all subsequent rollouts leading to poor update steps of the agent. In our experiments, no algorithm tested managed to escape from these dead-lock situation when trained from scratch. Thus, we applied a curriculum which slightly increases the scrap costs of a component scraped at one of the assemblies. The intention behind is that first, the agents learn the general mechanics of the production line and which actions are necessary to bring parts to the final station. Then, they have to get more efficient by reducing the number of scrap parts. We increase the scrap weight by a constant factor until $\frac{1}{k}$ is reached (in our experiments 0.006 for the scenario having $k = 3$) once the reward is above 100 in five subsequent evaluations. As optimizing CL requires to memorize past actions, we consider only two algorithms here: A stacked PPO (Schulman et al., 2017) receiving the past $k$ observations (with $k = 40$ and $k = 100$) and a recurrent PPO (Heess et al., 2015) as in (Pleines et al., 2023). First, we can see that using an curriculum is *effective*: Agents obtained through a curriculum obtain a large positive reward right from the beginning and they manage to keep it also when the scrap costs are increased. Figure 8 shows a comparison of the recurrent PPO trained with and without the curriculum for different random seeds and learning rates. In addition, we see that the stacked version of PPO does not manage to reach the performance of the recurrent version, even when trained 10 times longer.

When implementing a baseline for CL, it is essential to consider the waiting time of the source as well as the distribution of components and workers. Our heuristics prioritized the buffer with the lowest fill level fills, while buffers feeding into later assembly stages receive higher priority. We then conducted a grid search over various waiting times and worker distributions. The best heuristic we identified achieved a reward of $254.6 \pm 1.1$. Overall, even the best-performing RL algorithms in our CL benchmark still fall short of the manually implemented heuristic.

## 6. Discussion

In this work, we explore a rich problem class from manufacturing and make it accessible for RL research by introducing a novel framework LineFlow. We demonstrated that RL algorithms can learn effective control policies for various production line challenges, achieving optimal performance in well-understood scenarios. Moreover, we showed that for more complex, dynamic production settings, traditional RL approaches struggle without additional techniques such as curriculum learning and memory-based policies. We found that learning effective control strategies from interactions alone is difficult as the ramp-up phase of production plays a crucial role in training stability and poor early decisions can lead to deadlocks and stalled learning. Thus, more complex settings require structured learning approaches, including curriculum learning and hierarchical control.

Researchers and engineers can now use LineFlow to implement their specific production settings and systematically generate simulation data. It enables both the testing of custom policies and the training of RL agents. The behavior of trained agents can be used for analysis, but these policies can also be deployed in the real-world. LineFlow enables research in diverse directions. It supports the study of non-stationary problems, such as machine breakdowns or processing time drifts, which are common in real-world manufacturing. Additionally, improving reward structures for curriculum learning and advancing algorithms could significantly reduce training times and enhance performance in complex production settings. Another promising avenue is transfer learning, which could facilitate knowledge transfer across related tasks for more efficient and effective solutions. Beyond RL, LineFlow serves as a data generator, enabling analysis of production line dynamics, bottleneck prediction, and maintenance forecasting through supervised learning. To advance RL research in production control, a unified training and evaluation framework is essential for tracking and improving the state of the art. LineFlow fills this gap and drives further research into optimizing agent performance in complex, dynamic manufacturing environments.

## Acknowledgments

The project *LineFlow* is funded by the Bavarian state ministry of research. TW is funded by the *Hightech Agenda Bavaria*. We thank our research partners from Robert Bosch GmbH and DMG Mori AG for helpful discussions, particularly Martin Roth, Dominik Böhnlein, Markus Guggemoos, and Thomas Stark. Many thanks to Matthias Burkhardt, Cindy Buhl, Kilian Führer, Andreas Fritz, Fabian Hueber, Lea Müller, and Edgar Wolf for their helpful suggestions.

## Impact Statement

This work introduces a reinforcement learning framework for active control of production lines, enabling optimized operations with improved efficiency. By addressing the challenges of real-time decision-making in dynamic manufacturing environments, our method supports sustainable industrial practices by using existing resources of manufacturing lines in an optimal way. This research aligns with global efforts toward smarter and greener production systems, benefiting industries, workers, and society.

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

# A. An Example Use Case in LineFlow

To demonstrate how LineFlow can be used to model and simulate production lines, and let an agent interact with it, we provide a step-by-step implementation of a simple production line.

## A.1. Implementing the Layout

In LineFlow, a production line is represented by a class that extends the `Line` class. Inside the `Line.build` method, we define and connect the various stations, setting their processing times and capacities. Below is an example of a small production line, which consists of two source stations, one supplying the main component and the other providing an additional component required for assembly. These components are fed into an assembly station, which combines them into a single unit. The assembled product then passes through a process station. After processing, a switch directs the product to one of two possible output paths, each leading to a sink station that collects the finished items.

```python
from lineflow.simulation import Line, Source, Sink, Assembly, Process, Switch

class ShowCase(Line):

    def build(self):

        source_main = Source(
            'Source1',
            processing_time=5,
            carrier_capacity=2,
            actionable_waiting_time=True,
        )
        source_comp = Source(
            'Source2',
            processing_time=5,
            part_specs=[{"assembly_condition": 100}],
            actionable_waiting_time=True,
        )
        assembly = Assembly('Assembly', processing_time=40, NOK_part_error_time=5)

        process = Process('Process', processing_time=15)
        switch = Switch('Switch', processing_time=1)
        sink_1 = Sink('Sink1', processing_time=70)
        sink_2 = Sink('Sink2', processing_time=70)

        assembly.connect_to_component_input(station=source_comp, capacity=2, transition_time=5)
        assembly.connect_to_input(source_main, capacity=3, transition_time=5)
        process.connect_to_input(assembly, capacity=2, transition_time=2)
        switch.connect_to_input(process, capacity=2, transition_time=2)
        sink_1.connect_to_input(switch, capacity=3, transition_time=2)
        sink_2.connect_to_input(switch, capacity=3, transition_time=2)
```

## A.2. Interaction with Agents

In LineFlow, agents interact with the production line by receiving observations from the environment and taking actions to optimize the system's performance. The state space includes features such as buffer fill levels, processing times, and the number of workers assigned to a station. Based on these observations, the agent can adjust parameters like switch routing decisions, worker assignments, or waiting times at sources. Agents can of course be trained, but also defined as a function. The following agent dynamically adjusts the output buffer selection at the switch based on the minimum buffer fill level.

```python
def agent(state, env):
    fills = np.array([state[f'Buffer_Switch_to_Sink{i}']['fill'].value for i in [1, 2]])
    return {
        "Switch": {"index_buffer_out": fills.argmin()}
        "Source1": {"waiting_time": 5},
        "Source2": {"waiting_time": 2},
    }
```

To observe the simulation in action, we enable visualization by setting `realtime=True`. This ensures that the system updates in real-time, allowing for a clear understanding of how products move through the production line. The simulation speed can be adjusted using the `factor` parameter, which controls the scaling of time units. For instance, setting `factor=0.1` speeds up the process by a factor of ten, making it easier to analyze system behavior over extended periods

```
line = ShowCase(realtime=True, factor=0.1)
line.run(simulation_end=150, agent=agent, visualize=True)
```

### A.3. Analysing the Data

The user can get all the simulation data in form of a pandas dataframe, consisting of the states of all `lineobjects` using `line.get_observations()`. LineFlow also provides preinstall analysis for basic performance indicators like `line.get_n_parts_produced()` and `line.get_n_scrap_parts()`. A sample visualization is shown in Figure 9.

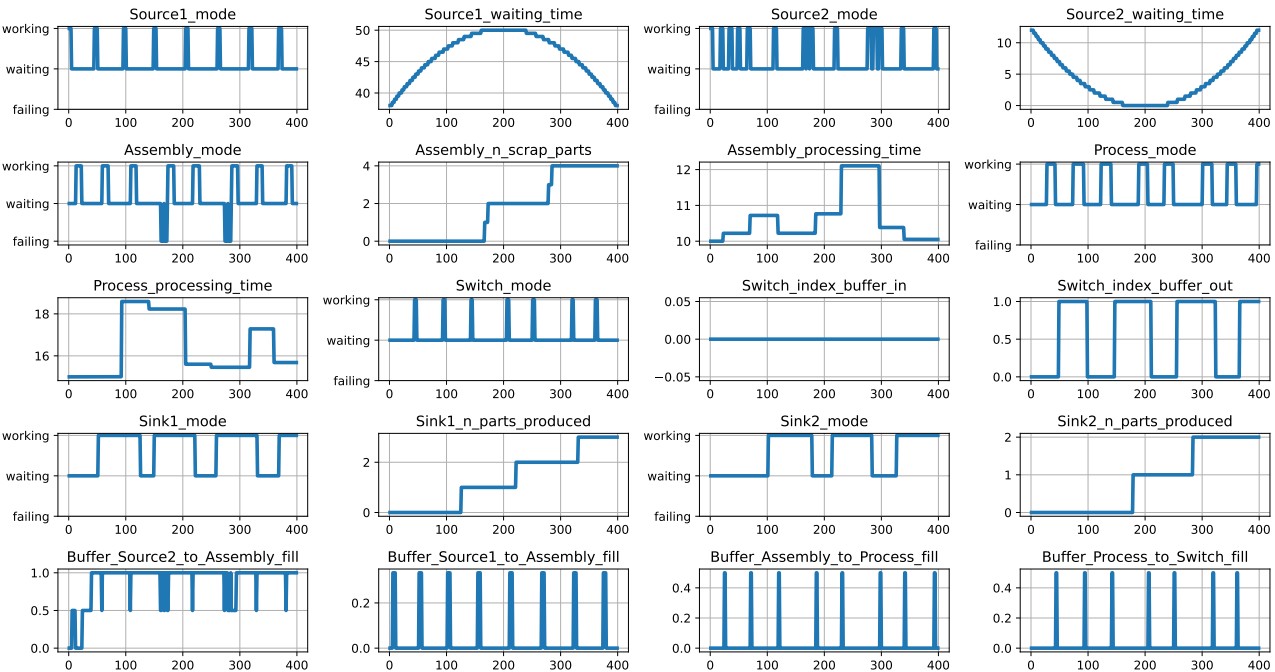

*Figure 9.* Selection of features extracted from the line displayed in Figure 2 visualized over time.

### A.4. Training RL Agents

To train reinforcement learning (RL) agents, we implemented the `LineSimulation(gym.Env)` class, which takes a `Line` object and a simulation duration as initialization parameters. This environment integrates seamlessly with `stable_baselines3` (Raffin et al., 2021), allowing RL models to interact with the simulation by specifying it via the `env` parameter. The following example demonstrates how to train an RL agent using PPO:

```
from stable_baselines3 import PPO

line = ShowCase()
env = LineSimulation(line, simulation_time=4000)
model = PPO("MlpPolicy", env, verbose=1)
model.learn(total_timesteps=100_000)
```

### A.5. General Overview

LineFlow is implemented in an object-oriented fashion. The central object is the `Line` object, which is decomposed of multiple `LineObjects` (Section A.6). A `LineObject` can either be *stationary*, like an assembly, or *movable*,

like a carrier. Each `LineObject` has a `ObjectState` representing parameters of the system a parameter. Each `ObjectState` consists of atomic states where each can be *actionable* or *non-actionable* and *observable* or *non-observable* (see Section A.7). The states of all objects in a line are accumulated in the `LineState` which provides a centralized interface for `Agents` interacting with the `Line` during simulation.

### A.6. Different Types of line objects available in LineFlow.

Typical properties when instantiating stationary objects are $T$ and $S$ that define their processing time $\mathcal{T} = T + \mathrm{Exp}_S$ as well as a *rework probability*, which models rework by letting a process run multiple times in a row. Any stationary object must be connected to other objects. This is typically done via one or multiple `Buffer` objects. For instance, a `Switch` can have arbitrarily many incoming and outgoing buffers where an `Assembly` must have exactly one incoming and one outgoing for the main carrier and arbitrarily many incoming for components that need to be joined with the carrier from the main track.

| Name | | Type | Description |
|------|---|------|-------------|
| Process |  | stationary | A station that simulates a processing step on the part. The processing step can be set to repeat due to a simulated human error, which doubles the processing time. This leads to a doubled processing time. |
| Sink |  | stationary | Removes components from a carrier. Carriers arrived here are marked as **OK**. Empty carrier either removed or returned to a `Magazine` or `Source` using a separate out-buffer depending on the layout. This way, the station can be used in both linear and circular lines. |
| Source |  | stationary | Places parts onto carriers. Can set individual properties, called `PartSpec`, to every part set up, like the assembly condition $T_{\mathrm{AC}}$. Carriers are either created, taken from a `Magazine`, or fetched separate incoming buffer. |
| Assembly |  | stationary | A Station for simulating assembly activities on the line. Individual parts and components are delivered with individual carriers, assembled with a simulated processing time and forwarded to the downstream station. Can be connected to a `WorkerPool` that can assign `Worker` objects to it modifying its processing time. |
| WorkerPool |  | stationary | Holds a predefined set of `Worker` objects and is attached to a fixed number of stations. Multiple pools can coexist for a production line allowing to modelling different skills or experience of workers. |
| Magazine |  | stationary | Magazine station is used to manage the carriers. The total number of carriers available to the line can be controlled via this station. The capacity of the carriers, i.e. the possible number of components that can be added at the assembly station, is also determined by this station. If the number of carriers is not of interest, the source can place the parts directly on carriers and no magazine is required. |
| Switch |  | stationary | The Switch distributes carriers to different stations, enabling parallel structures within the line. |
| Buffer |  | stationary | The Buffer transports carriers from one station to another. Time needed to push and get carriers to and from can be specified as well as its *capacity* and time a carrier needs to traverse the buffer. |
| Carrier |  | movable | Is set up at a `Source` station or a `Magazine` and holds a predefined number of `Part` objects. |

| Part | | movable | Single unit which is initially created at a `Source`. Holds a `PartSpec` each station handling it can access and individually adapt to. |
| Worker | | movable | Belongs to a `WorkerPool` and can be assigned to a station. Traversal time can be configured. |

Table 2: Different types of `LineObjects`.

## A.7. Different Types of `States`

This section provides an overview over the elementary states implemented in LineFlow. Every state is associated to at least one `LineObject` and the mapping of selected objects to states is given in Table 5. In general, a state can be *actionable* or *non-actionable* meaning that agents can set them to other values within their value range via runtime. Per default, every state is *observable*, meaning that they are part of the observational space agents have access to. However, to facilitate ablation studies or simulate sensor failures, any state can be selectively masked as *non-observable*. Many states are consequences of events taking place at the line, like how many carriers are on a buffer, and cannot be influenced directly. All states kept updated constantly and represent at any time the current situation. Some states, however, are lagged in the sense that their update happens once an certain event is triggered. For instance, the state holding the processing time of a station is updated once the process is over with the value of the last process.

The states can be categorized into two main categories: *discrete* and *numeric*. Discrete States on the one side, handle categorical items such as station modes. Internally, the values of a discrete state are labeled encoded to integer numbers. A subclass are *count* states that represent integer numbers, like the number of scrap parts at a station. Numeric states, on the other side, hold continuous numerical values, like processing times or buffer fill levels. Their value range may be restricted by upper and lower tolerances. Table 3 provides a list of states implemented in LineFlow. All states are mapped to the respective states of gymnasium (Towers et al., 2024). At any simulation step $t$, the states of all `LineObjects` are fetched and put into an agent, which returns new values for the list of actionable features to be updated accordingly.

| Name | Type | Actionable | Description |
|---|---|---|---|
| Mode | discrete | non-actionable | The mode a station is currently in. Is either `working`, `failing`, or `waiting`. |
| Processing time | numeric | non-actionable | The processing time of the last process. |
| Waiting time | numeric | actionable | Value used to wait till two parts produced. |
| Station-Assignment | discrete | actionable | Discrete value denoting to which station a worker is assigned to. Must be one of the predefined stations attached to the worker pool the worker belongs to. |
| #Workers | count | non-actionable | Number of workers assigned to a station. Aggregated feature computed from the station-assignment of all workers |
| #OKs and #NOKs | count | non-actionable | Number of OK and NOK parts the station has produced so far. |
| Buffer-Indices | discrete | actionable | The incoming and outgoing index of a stationary object, like a `Switch`, it should get and push components to. |
| Buffer-Fill | numeric | non-actionable | Relative number of carriers on a buffer. |

Table 3: Examples of states.

| | Processing time | Current mode | Number of workers | $n_{ok}$ | Waiting time | In and out buffer |
|---|---|---|---|---|---|---|
| Assembly | O, N-A | O, N-A | O, A | O, N-A | - | - |
| Process | O, N-A | O, N-A | O, A | O, N-A | - | - |
| Source | O, N-A | O, N-A | - | O, N-A | O, A | - |
| Switch | O, N-A | O, N-A | - | - | - | O, A |

*Table 5.* Default mapping of selected stationary objects to states. Letters *O* and *A* stand for observable and actionable respectively, where *N* marks *non-observable* and *non-actionable*.

### A.8. Visualization

Once a layout is implemented in LineFlow, an object of the respective `Line` object allows to set `visualize=True` during a simulation run, which renders the current state in pygame. Figure 10. As layouts are implemented in custom line classes, users have control over how these lines are instantiated and can use the power of the programming language of python, i.e., parametrizing the number of sinks or assemblies via constructor arguments or using for-loops to set them up.

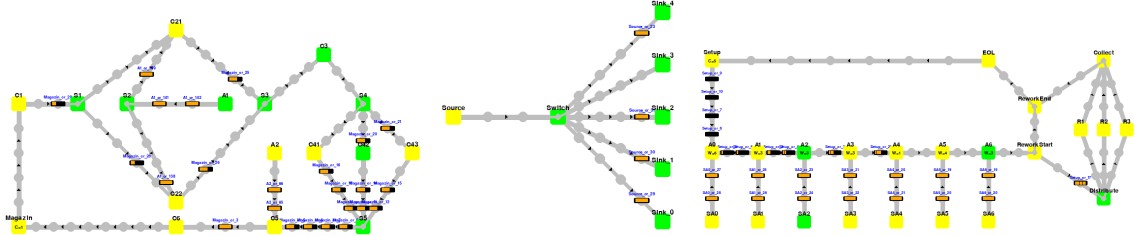

*Figure 10.* Some layouts implemented in LineFlow.

## B. Optimality Proofs for Case Studies

In this section, we give proofs for optimal control policies for the scenarios WT and WTJ in Section B.1, PD in Section B.2, and WA in Section B.3. In Section B.4, we state a heuristic for CL yielding near-optimal reward.

### B.1. Waiting Time

Let $\mathcal{T}_{S_C}$ and $\mathcal{T}_A$ be the process time distributions of $S_C$ and $A$ respectively and denote by $\mathcal{T}_p$ and $\mathcal{T}_g$ the time distributions to push and get a component to and from a buffer respectively. First, we state a formula for the optimal waiting time at $S_C$. Essentially, the waiting time has to fill the gap between the times $A$ and $S_C$ need to handle and process their parts, which are $\mathcal{T}_A + 2 \cdot \mathcal{T}_g + \mathcal{T}_p$ and $\mathcal{T}_{S_C} + \mathcal{T}_p$. Thus, the optimal waiting time can be globally determined as the expected value of their difference

$$T_W^* = \mathbb{E}[\mathcal{T}_A + 2 \cdot \mathcal{T}_g - \mathcal{T}_{S_C}]. \tag{3}$$

Particularly, it suffices in WT to learn a *static* waiting time. See also Figure 11 for how the waiting time at $S_C$ affects the reward. To calculate the maximum number of parts that can be produced, we have to consider the time that it take for the assembly to start processing as well as the time the last part produced in $T_{\text{sim}}$ needs to be finished by the sink. Let $\mathcal{T}_{S_C}, \mathcal{T}_{S_M}$, $\mathcal{T}_A$, and $\mathcal{T}_S$ be the process time distributions of $S_C$, $S_M$, $A$, and the sink $S$ respectively. Moreover, let $T_{S_C \to A}, T_{S_M \to A}$, and $T_{A \to S}$ be the times a part needs to traverse on the buffers between $S_C$ and $A$, $S_M$ and $A$, and $A$ and the sink $S$. The assembly can start its processing once a part from $S_M$ and $S_C$ has arrived, that is

$$\mathcal{T}_{\to A} := \max\{\mathcal{T}_{S_C} + \mathcal{T}_p + T_{S_C \to A}, \mathcal{T}_{S_M} + \mathcal{T}_p + T_{S_M \to A}\}$$

where $\mathcal{T}_p$ and $\mathcal{T}_g$ denote the times to push and get a component to and from a buffer respectively. As the first part also needs to transfer to the sink $S$ and has to be processed it, the expected value of the maximal parts produced from 0 to $T_{\mathrm{sim}}$ is:

$$\mathbb{E}\left[\frac{T_{\mathrm{sim}} - \mathcal{T}_{\rightarrow A} - T_{A \rightarrow S} - \mathcal{T}_g - \mathcal{T}_S}{\mathcal{T}_A + 2 \cdot \mathcal{T}_g + \mathcal{T}_p}\right]. \tag{4}$$

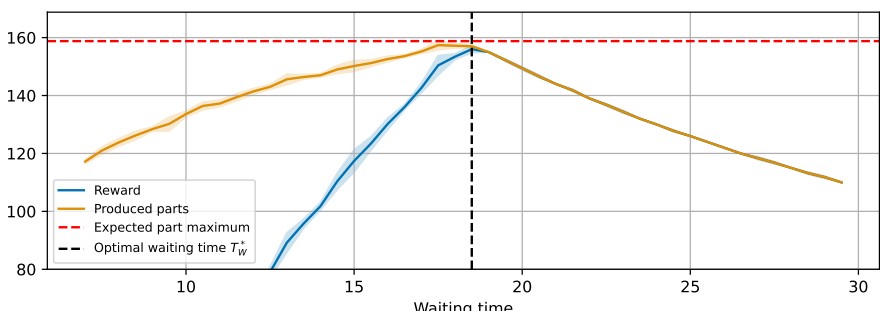

*Figure 11.* Visualisation of the setting in WT with optimal waiting time $T_W^* = 18.5$ as in Equation (3) and the maximal number of expected parts as computed in Equation (4).

Next, we explain the construction of $f$ and $T_{\mathrm{jump}}$ of $\mathrm{WT}_J$ with a jumping processing time of the assembly $A$. Following Equation (4) and ignoring the ramp-up and jumps of the assembly station, the expected number of parts $N$ produced in $T_{\mathrm{sim}}$ is

$$N = \frac{T_{\mathrm{sim}}}{\mathbb{E}[\mathcal{T}_A + 2\mathcal{T}_g + \mathcal{T}_p]} = \frac{T_{\mathrm{sim}}}{T + S + E}$$

with $E = \mathbb{E}[2\mathcal{T}_g + \mathcal{T}_p]$. When the assembly processing with $f \cdot T + \mathrm{Exp}_S$ instead of $T + \mathrm{Exp}_S$ for a period of $T_{\mathrm{jump}}$, the number of expected parts is:

$$\frac{T_{\mathrm{sim}} - T_{\mathrm{jump}}}{T + S + E} + \frac{T_{\mathrm{jump}}}{fT + S + E}$$

Clearly, if $f > 1$, the expected number of produced parts must be smaller than $N$. Thus, we want to construct $f$ from a sampled $T_{\mathrm{jump}}$ such that the expected number of produced parts is $R \cdot N$ for a fixed constant $0 < R < 1$. Then, for a for a sampled $T_{\mathrm{jump}}$, we look for $f$ such that the expected parts satisfies:

$$\frac{T_{\mathrm{sim}} - T_{\mathrm{jump}}}{T + S + E} + \frac{T_{\mathrm{jump}}}{fT + S + E} = R \cdot N$$

It is not hard to see that this is the case for

$$f = \frac{1}{T}\left(\frac{T_{\mathrm{jump}} \cdot (T + S + E)}{(R - 1) \cdot T_{\mathrm{sim}} + T_{\mathrm{jump}}} - S - E\right).$$

To verify that the maximal number of produced parts is in fact $R \cdot N$, we run simulations in LineFlow using varying values of $R$ and a fixed waiting time of 0 at the source to keep the assembly constantly loaded. As this would lead to scrap, which in turn blocks the assembly time for a certain amount of time, we additionally set a very large assembly condition $T_{AC}$. Figure 12 shows the empirical frequency of the number of parts produced for WTJ for different $R$ values as well as $R \cdot N$ with $N$ the expected value of WT showing that $f$ and $T_{\mathrm{jump}}$ are constructed as desired.

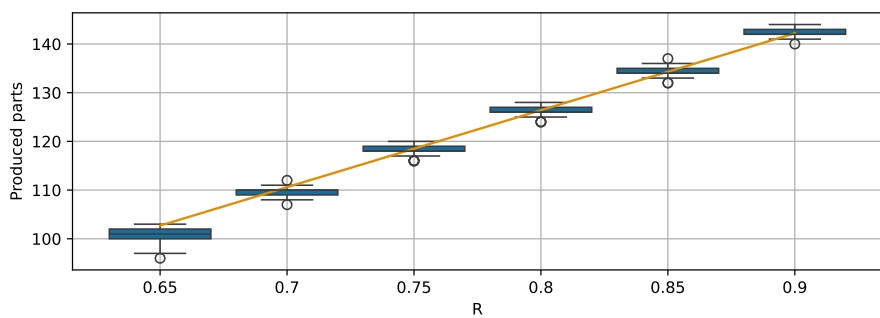

*Figure 12.* Number of parts produced for WTJ and WT with large assembly condition $T_{\text{AC}}$ and no waiting at $S_C$ for varying $R$.

To get an optimal policy for WTJ, we estimate $\mathbb{E}[\mathcal{T}_A + 2 \cdot \mathcal{T}_g - \mathcal{T}_{S_C}]$ from Equation (3) by regressing on $\mathbb{E}[\mathcal{T}_A]$ with the processing times reported from $A$. Essentially, we take a rolling mean of the last $l$ processing times observations from $A$. Clearly, the larger $l$, the better the estimate for $\mathbb{E}[\mathcal{T}_A]$ in a time period where the mean does not jump. However, the larger $l$, the worse the new estimate of the waiting time adjusts to a new level. By varying $l$ and testing the agent for WTJ (see Figure 13), we found that $l = 1$ gives the best reward. This reward is used as optimal value in Section 5.

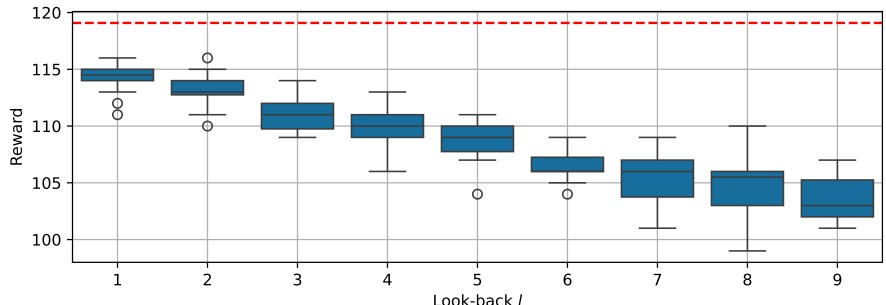

*Figure 13.* Reward for agent for varying look-back $l$.

## B.2. Part Distribution

In Section 4.2, we showed that the maximal number of parts that can be produced in $\text{PD}_k$ is

$$\mathbb{E}[\mathcal{N}] := \sum_{i=1}^{k} \frac{T_{\text{sim}}}{(1 + S_i) \cdot T_i} \tag{5}$$

Particularly, the fraction of all parts produced by $P_i$ is

$$\rho_i = \mathbb{E}\left[\frac{\mathcal{N}_i}{\mathcal{N}}\right] = \frac{1}{\sum_{j=1}^{k} \frac{T_i}{T_j}} \tag{6}$$

and consequently, the optimal distribution policy for the switch also needs to deliver that number to $P_i$ to reach the maximal number of parts. Put differently, the optimal policy sends a part with probability $\rho_i$ to $P_i$. We have implemented in LineFlow an agent that greedily controls the switches: A carrier is pushed on a buffer having lowest fill and fetched from a buffer with highest fill. This greedy policy have been evaluated for $k \in \{3, 4, 5\}$ and its part distribution is compared to the optimal distribution $(\rho_1, \ldots, \rho_k)$ from Equation (6). The result is shown in Figure 14.

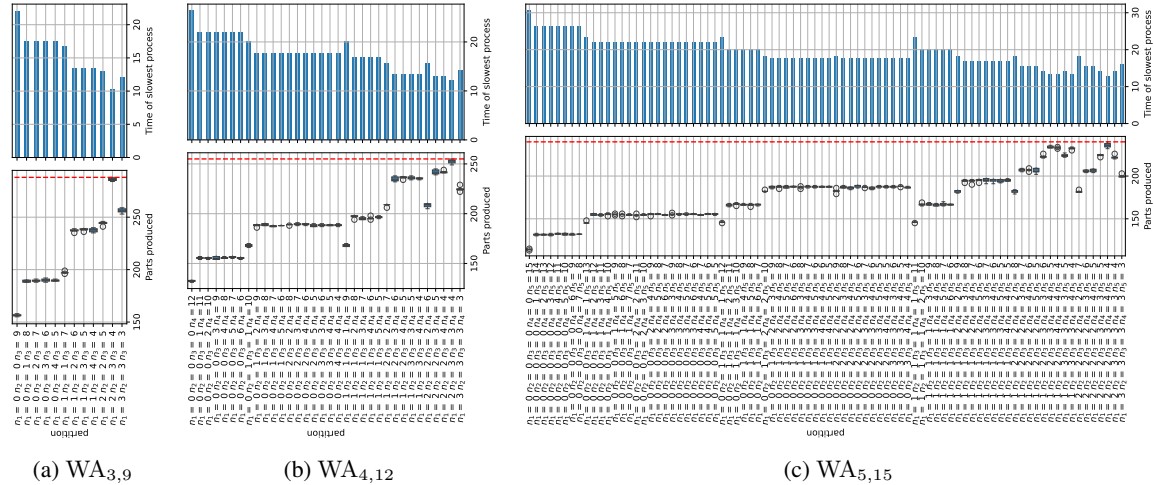

(a) WA$_{3,9}$ (b) WA$_{4,12}$ (c) WA$_{5,15}$

*Figure 15.* Performance of all possible monotone partitions for $T_i = (16 + i \cdot 4)$, and $c = 0.3$ for a simulation length of 2000 in the worker distribution example.

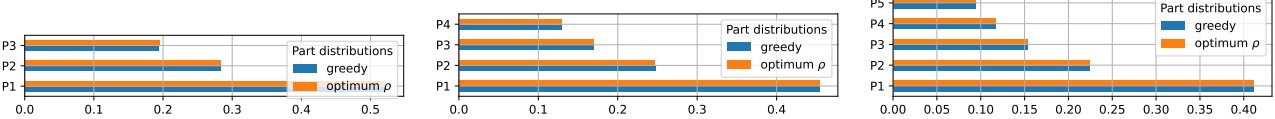

*Figure 14.* Part distributions of the greedy policy compared with optimal distribution for $k \in \{3, 4, 5\}$.

## B.3. Worker Assignments

In this section, we formulate

$$T_C^* = \min_{(n_1,\ldots,n_k) \in \mathcal{N}_{N,k}} \max_{i \in [k]} \mathbb{E}[\mathcal{T}_{T_i,S_i,n_i}]$$

as given in Equation (2) as integer optimization problem. Recall that $\mathbb{E}[\mathcal{T}_{T_i,S_i,n_i}] = T_i \cdot (p_c(n_i) + S_i)$. We equivalently reformulate the minimization problem by introducing a real-valued auxiliary variable $m$ as follows:

$$\min_{n_1,\ldots,n_k,m} m$$
$$\text{subject to} \quad m \geq T_i \cdot (p_c(n_i) + S_i) \quad \forall i \in [k]$$
$$\sum_{i=1}^{k} n_i = N \tag{7}$$
$$n_i \in \mathbb{N}_{\geq 0} \quad \forall i \in [k]$$
$$m \in \mathbb{R}_{\geq 0}$$

In our benchmarks, we used $T_i = (16 + i \cdot 4)$ and $c = 0.3$ for varying $k$ and $N = 3 \cdot k$. We implemented this optimization problem in gekko (Beal et al., 2018). Solving Equation (7) yields $(2, 3, 4)$ for $k = 3$, $(2, 3, 3, 4)$ for $k = 4$, and $(2, 2, 3, 4, 4)$ for $k = 5$. To evaluate the correctness of LineFlow, we enumerated and evaluated all monotonic worker assignment, that is $n_i \leq n_j$ for $i \leq j$, for $k \in \{3, 4, 5\}$ for multiple runs with simulation length 2000 (see Figure 15). We found the optimal assignment obtained empirically matches the exact optimum obtained by using Equation 7.

| Name | Value |
|---|---|
| Simulation time $T_{\text{sim}}$ | 4,000 |
| Step size $T_{\text{step}}$ | 1 |
| Rollout steps | 1,000 |
| Discount factor $\gamma$ | 0.99 |
| Number of environments (vectorized) | 5 |
| Number of stacked observations | 1 |
| Advantage normalization (only PPO) | False |
| Batch size in update | 1,000 |
| Number of epochs in update | 5 |
| Clip range (only PPO and A2C) | 0.2 |
| Maximal gradient norm (only PPO) | 0.5 |
| Policy (stable-baselines3) | `MlpPolicy` (`MlpLstmPolicy` for recurrent PPO) |

*Table 7.* General hyperparameters used for all algorithms.

### B.4. Complex Line

We encountered several challenges when designing an effective control policy for the complex line. Due to the small buffer capacities between assembly stages, the line is prone to blockages, leading to scrap parts and a significant drop in reward. Maintaining a steady production flow proved to be crucial. To resolve potential jams as efficiently as possible, we implemented a switch distribution policy that prioritizes later assembly stations. While this approach helped mitigate blockages, it also introduced new challenges, such as imbalanced utilization of early-stage buffers, which could lead to delays in part availability. Regarding worker distribution, our analysis showed that keeping a fixed distribution throughout the episode consistently outperformed any of our redistribution strategies. We attribute this effect to the time required for worker redistribution, which likely disrupts the system's stability. Additionally, frequent reassignments may introduce inefficiencies due to the time needed for workers to relocate and adapt to new tasks. We also explored adaptive worker allocation strategies that dynamically reassign workers based on buffer fill levels. However, these approaches often led to oscillatory behavior, where workers continuously moved between stations without improving throughput. This suggests that the overhead of frequent redistribution outweighs its potential benefits in our setup.

## C. Benchmarking Details

In our experiments, we used stable-baselines3 (Version 2.3.2). In total, 22 models have been trained and evaluated for every scenario, each model using three different random seeds resulting in 66 trained models for each scenario. The runtimes vary: Roughly 14 hours for $PD_5$ are required to train a single model, whereas only 7 hours for WTJ and 2 hours for WT. To train a stacked and recurrent version of PPO for scenario CL NVIDIA H100 GPUs were used. A single stacked PPO model trains roughly 20 hours, where the recurrent models train for almost 6 days.

## C.1. Waiting Time

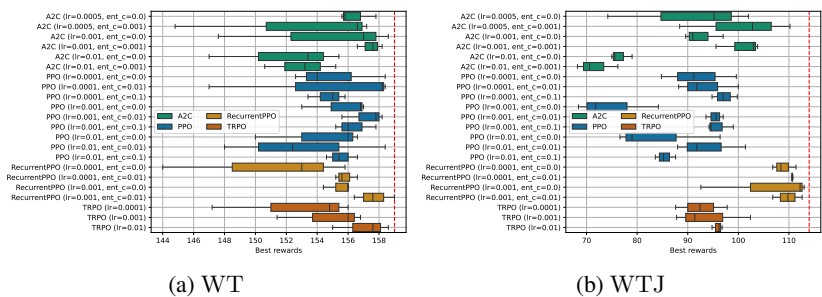

(a) WT              (b) WTJ

*Figure 16.* Best performance of algorithms on evaluation environments for WT and WTJ.

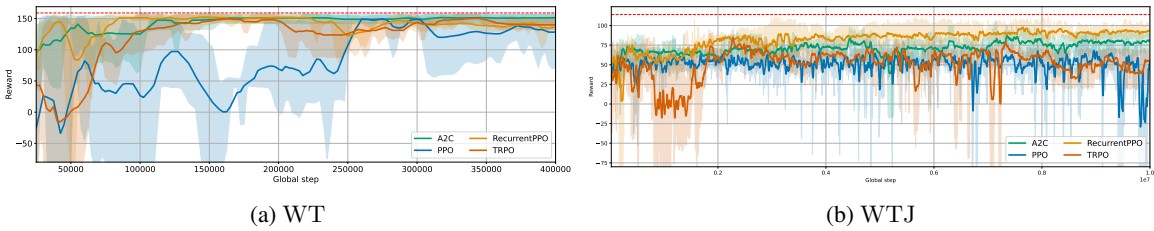

(a) WT              (b) WTJ

*Figure 17.* Reward over steps for WT and WTJ.

## C.2. Part Distribution

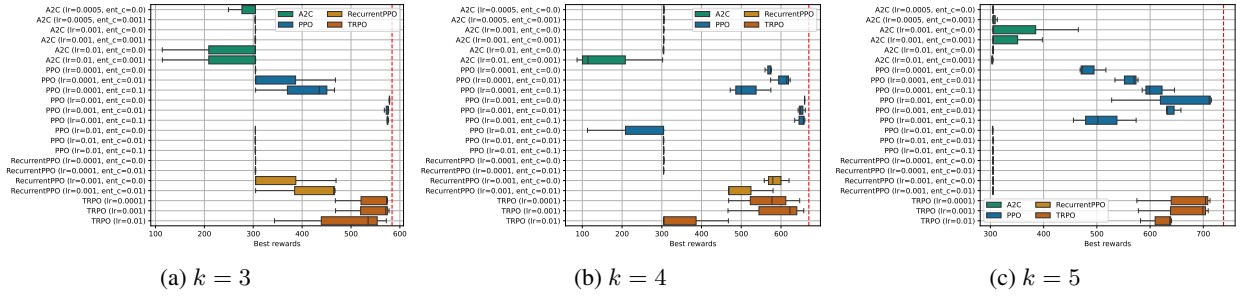

(a) $k = 3$       (b) $k = 4$       (c) $k = 5$

*Figure 18.* Best performance of algorithms on evaluation environments for $\mathrm{PD}_k$.

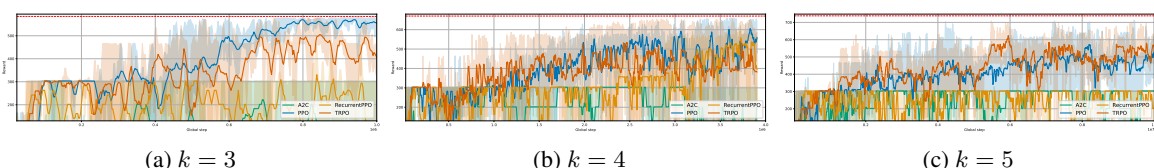

(a) $k = 3$       (b) $k = 4$       (c) $k = 5$

*Figure 19.* Reward over steps for $\mathrm{PD}_k$.

## C.3. Worker Assignment

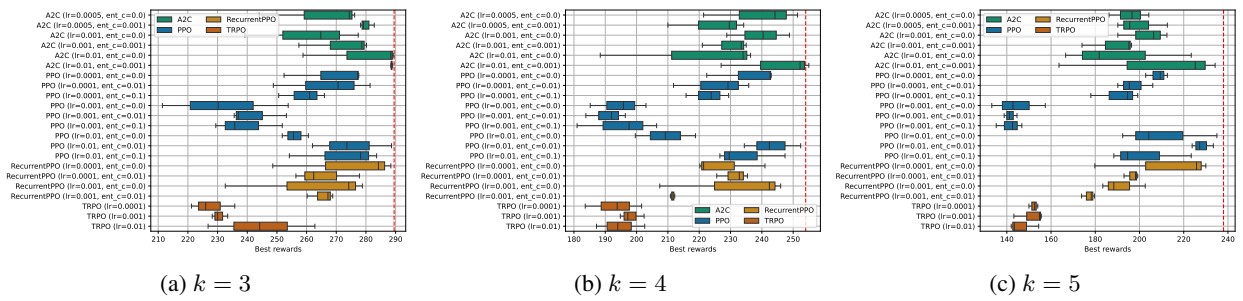

(a) $k = 3$      (b) $k = 4$      (c) $k = 5$

*Figure 20.* Best performance of algorithms on evaluation environments for $WA_{k,3k}$.

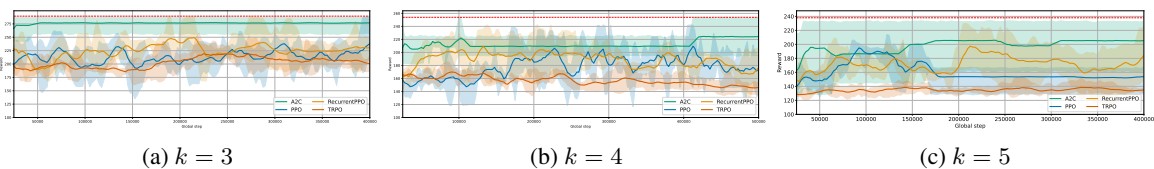

(a) $k = 3$      (b) $k = 4$      (c) $k = 5$

*Figure 21.* Reward over steps for $WA_{k,3k}$.

## D. Validation on Real Production Data

In this section, we evaluate the sim-to-real gap of LineFlow using the publicly available production line dataset from (Risdal et al., 2016). In this dataset, each row corresponds to a produced part, and the columns record timestamped feature activations at various stations. Based on this information, we reverse-engineered the production layout, which consists of two parallel pre-assembly lines with 12 stations each (see Figure 23), both feeding into a shared subsequent line. For our analysis, we focused on the pre-assembly line responsible for the majority of component production.

We first analyzed the distribution of processing times at each station and found that they closely follow exponential distributions, which supports the modeling assumptions used in LineFlow (see Figure 22). The limited resolution of the provided timestamps required a pooling of the processing times of successive parts. Specifically, because individual job durations could not be precisely resolved, we applied a rolling, non-overlapping window over the production sequence and computed the average processing time across 100 consecutive parts. This smoothing technique allowed us to approximate the underlying processing time distribution while mitigating the effects of timestamp granularity.

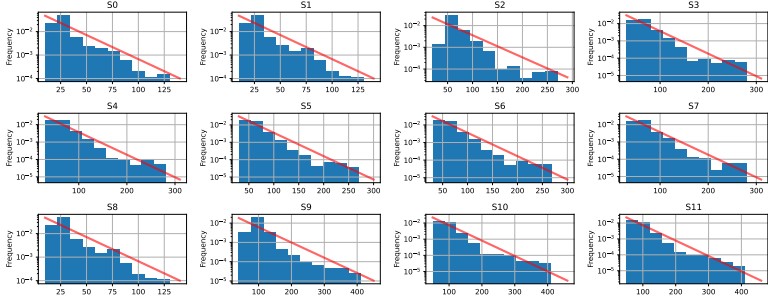

*Figure 22.* Histograms of the averaged processing times in a log scale together with a fitted exponential distribution in red.

Next, we implemented the reconstructed layout in LineFlow and with the aim to compare the number of parts produced in the simulation with those observed in the real system. Since the dataset does not provide information about traversal

times or the number of slots in the connecting buffer, we made the simplifying assumption that traversal times are negligible compared to station processing times. Additionally, we assumed that the switches before and after the stations distribute parts in a round-robin fashion. The results showed a close match (see Figure 23): The number of parts produced by our simulation matches the number of parts produced by the real system, providing empirical evidence that it accurately models real-world production dynamics.

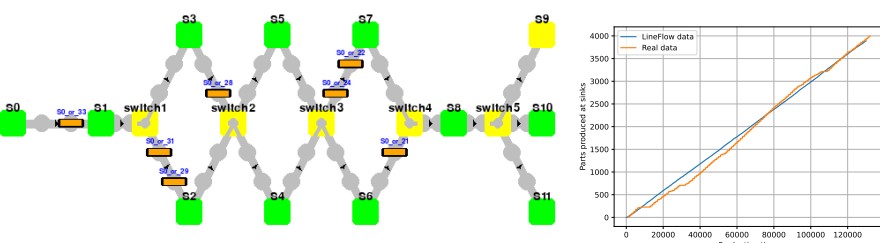

*Figure 23.* The implemented layout in LineFlow (left) and its simulated output compared with the output of the real production line (right).

