# OpenReview forum: "LineFlow: A Framework to Learn Active Control of Production Lines"
_ICML.cc/2025/Conference — ICML 2025 poster_

### Official Review · Reviewer_9j5R · 2025-03-11

**Overall Recommendation:** 3

**Summary:**

Reinforcement learning (RL) has demonstrated potential in optimizing production line control. However, a standardized and general framework remains absent. To address this, the authors present LineFlow, an open-source, extensible Python framework for simulating production lines of arbitrary complexity and training RL agents to manage them.

**Claims And Evidence:**

The primary contribution of this submission is a Python framework for simulating production lines with RL agents, and this is supported by the code provided in the supplementary material.

**Essential References Not Discussed:**

I believe the optimality proof in Appendix C should be supported with relevant references.

**Experimental Designs Or Analyses:**

Although the authors design diverse case studies, this paper aims to introduce a new framework for simulating production lines. Therefore, it should include real-world case studies built using this framework.

**Methods And Evaluation Criteria:**

This paper does not introduce a new method but demonstrates various scenarios implemented using LineFlow.

**Other Comments Or Suggestions:**

Can the authors use this framework to simulate real-world applications?

**Other Strengths And Weaknesses:**

Despite the importance of active line control across various industries, no well-grounded simulation framework has been available for training RL agents in production line settings. Thus, this new framework has significant impact. However, since the experiments are limited to a few case studies, I am concerned about its generalizability to real-world applications.

**Questions For Authors:**

With the growing complexity of production lines involving numerous entities, a multi-agent approach is sometimes necessary. Is this framework extensible to multi-agent RL?

**Relation To Broader Scientific Literature:**

The environment in the proposed framework is well-designed, drawing from research on active control of production lines.

**Theoretical Claims:**

Table 1 presents a problematic result where the optimal reward for the WT case is lower than the reward achieved by RL algorithms, despite the provided optimality proof.

---

> ### Author Rebuttal · Authors · 2025-03-31
>
> We are excited that the reviewer
> finds that "Despite the importance of active line control across various industries, no
> well-grounded simulation framework has been available for training RL agents in production line
> settings. Thus, this new framework has significant impact".  We thank the reviewer for the suggestions.
>
> ## Real-World validation
> A key concern raised by the reviewer was whether LineFlow is capable of accurately simulating real-world applications.
> We would like to clarify that the case studies presented in the
> paper — WT (waiting time), WTJ (waiting time with jumps), PD (part distribution), and WA (worker
> assignment) — are not synthetic toy problems but are, in fact, directly motivated by real-world
> challenges faced by our industrial partners.
> While we abstracted and isolated these problems to enable mathematical analysis and controlled
> benchmarking, they reflect real production phenomena observed in practice. They
> are furthermore reflected in the literature, as  the part/assembly line feeding problems [2, 3, 4, 5, 6].
>
> Nevertheless, we agree with the reviewer that a real-world validation can strengthen the impact of
> our work. Generally, there exist strong privacy concerns among manufacturers when it comes to
> releasing production line layouts. Simply releasing the cycle time of a line alone can give a
> competitor information about production efficiency, capacity, and competitive advantage.
> Consequently, publicly available datasets holding performance information and non-trivial layouts
> are rare. Here, we hope that LineFlow will close this gap by allowing manufacturers to provide
> synthetic non-confidential digital twins preserving their key runtime challenges to the public.
>
> To further validate the sim-to-real gap of LineFlow, we conducted an evaluation using the publicly
> available Bosch production line dataset from [1]. We first analyzed the processing time
> distributions and confirmed that they closely follow exponential distributions, supporting the
> modeling assumptions used in LineFlow. Due to the insufficient resolution of the provided
> timestamps, this analysis was non-trivial.
>
> https://imgur.com/a/icml-rebuttle-images-LLTUpZF
>
> We then reverse-engineered the production layout (having 13 stations) from the dataset and
> implemented a corresponding simulation in LineFlow. The comparison between the number of parts
> produced in the simulation and the number observed in the real-world data showed a very close match:
> LineFlow produces $3909$ parts in a time frame $4000$ parts have been produced for the real dataset,
> providing strong empirical evidence that LineFlow accurately models real-world production dynamics.
> We will include this validation in an additional section in the appendix of the revised manuscript.
>
> Regarding industry implementations for active line control, we can only refer to our conversations with industry partners. These discussions revealed that active line control in production is often managed directly on the production line by personnel or through simple heuristics like the ones we implemented for our scenarios. Our results show that these heuristics perform well, but RL agents achieve comparable performance, demonstrating their viability as an alternative approach.
>
> ## Multi-agent RL
> The reviewer is right that multi-agent RL is a good tool for addressing the curse of dimensionality of
> large-scale assembly lines. In fact, we declared this as one of the future works.
>
> ### References:
> - [1] Meg Risdal et al., Bosch Production Line Performance. https://kaggle.com/competitions/bosch-production-line-performance, 2016. Kaggle.
> - [2] Daria Battini et al., The Integrated Assembly Line Balancing and Parts Feeding Problem with Ergonomics  Considerations, IFAC-PapersOnLine, Volume 49, Issue 12, 2016, Pages 191-196,
>   ISSN 2405-8963, https://doi.org/10.1016/j.ifacol.2016.07.594.
> - [3] Schmid, N. A., and Limère, V. (2019). A classification of tactical assembly line feeding problems. International Journal of Production Research, 57(24), 7586–7609. https://doi.org/10.1080/00207543.2019.1581957
> - [4] Linn I. Sennott et al.,
> Optimal Dynamic Assignment of a Flexible Worker on an Open Production Line with
> Specialists, European Journal of Operational Research, Volume 170, Issue 2, 2006, Pages 541-566, ISSN 0377-2217, https://doi.org/10.1016/j.ejor.2004.06.030.
> - [5] Chutima, P. and Chimrakhang, J. (2021), "Multi-objective worker allocation optimisation in a multiple U-line system", Assembly Automation, Vol. 41 No. 4, pp. 466-476. https://doi.org/10.1108/AA-12-2020-0198
> - [6] Ritt, M. et al., The assembly line worker assignment and balancing problem with stochastic worker availability. International Journal of Production Research, 54(3), 907–922. https://doi.org/10.1080/00207543.2015.1108534

---

### Official Review · Reviewer_R3X2 · 2025-03-12

**Overall Recommendation:** 2

**Summary:**

The paper introduces an open-source Python framework for simulating production lines and training RL agents to control them. The authors demonstrate the framework capabilities through several core subproblems of active line control with corresponding mathematical analyses. Results show that while RL policies approach optimal performance in simpler scenarios, complex industrial-scale production lines still present significant challenges, pointing to the need for further research.

**Claims And Evidence:**

* LineFlow effectively simulates production lines: The framework comprehensively models various elements of production lines including machines, buffers, and workers with sufficient flexibility and realism.
* RL agents can learn near-optimal policies for well-defined subproblems: Experiments demonstrate that RL algorithms approach optimal solutions for specific subproblems like optimal waiting time, part distribution, and worker assignment.
* Complex production line control requires additional techniques: For complex scenarios, basic RL approaches fail without curriculum learning, as shown in their "complex line" experiments.

**Essential References Not Discussed:**

Not to my knowledge.

**Experimental Designs Or Analyses:**

The experiments are competently designed but primarily demonstrate the application of known techniques to a specific domain rather than revealing fundamental new insights about RL or optimization algorithms.

**Methods And Evaluation Criteria:**

The methods and evaluation seems appropriate for the domain:
* The authors properly formulate production line control problems as MDPs, suitable for RL approaches.
* RL algorithms are benchmarked against theoretical optimal solutions and heuristic approaches, providing good comparisons.
Some limitations include relatively limited comparison with industry standard approaches and insufficient exploration of scalability to larger production systems.

**Other Comments Or Suggestions:**

No further comments.

**Other Strengths And Weaknesses:**

Strengths:
* Well written paper
* Addresses practical production line control problems
* Provides a comprehensive, well-designed framework
* Effective progression from simple to complex scenarios
* Thoughtful application of curriculum learning
Weaknesses:
* Limited real-world validation with actual production data
* Incomplete discussion of reward function design challenges
* Limited comparison with industry-standard approaches

**Questions For Authors:**

I might have missed this, but does LineFlow handle heterogeneous machine types as stated in the related work? Do your experiments simulation reflect that?

**Relation To Broader Scientific Literature:**

The work bridges production line optimization and RL but does not significantly advance either field. It represents a solid engineering effort rather than the kind of theoretical or algorithmic innovation expected at ICML. It might be an ideal paper for IEEE or other venues though!

**Theoretical Claims:**

The paper lacks novel theoretical contributions. The mathematical formulations presented as "theoretical foundations" are standard applications of existing optimization principles to specific production settings. These serve more as validation benchmarks than theoretical advances in either RL or production optimization.

---

> ### Author Rebuttal · Authors · 2025-03-31
>
> We appreciate the reviewer emphasizing that we provide "a comprehensive,
> well-designed framework" addressing "practical production line control problems"
> and "provide a good comparison of optimal solutions, heuristic approaches, and
> agents' scores". We would like to thank the reviewer for the insightful feedback and
> suggestions.
>
> ### Real-world validation with actual production data
>
> The same concern was raised by reviewer `9j5R`, and we want to point to the
> reply we gave there.
> Particularly, we extract processing and transition times from a real world production data set and
> show a comparison of real and simulated line performance. We will include these results in the
> appendix of the final version of our paper.
>
> ### Reward function design challenges
>
> We thank the reviewer for highlighting the need for a more complete discussion on reward function
> design. Common goals include maximizing parts built or revenue, with the widely used *OEE* score
> arising as a special case of our target score (equation 1). In general, LineFlow does not depend on
> a fixed reward function and the user can freely select any target goal to be optimized in the layout
> at hand.  To address this further, we will add a section detailing special cases of $C_\pi$ and
> their alignment with typical optimization objectives.
>
> ### Comparison with industry-standard approaches
>
> We are not completely sure what the reveiwer means with *industry-standard approaches* and in that
> particular context, *comparison*. If the reviewer refers to standard-approaches for *finding control
> policies* which need to be compared by their reached reward, we think a comparison is implicitly
> given already as we compare the control policies found by RL with the theoretically *best possible
> policy*. Thus, any other policy, whether standard or not, must be inferior to the RL solution. In
> practice, to the best of our knowledge, there exists no *standard approach* to find a good runtime
> control policy for a concrete layout, although there exist proprietary *software tools* that help to
> analyse production line and given control strategies. If, on the other hand, the reviewer means
> comparing the *data-efficiency*, this is a more involved question to answer, as this requires
> setting more rules on how much a priori information an approach is allowed to use. This is currently
> beyond the scope of our paper but definititely an intereseting future research question.
>
> We hope that adresses the question well enough and if not we hope the reviewer can elaborate the
> question a bit more. Please also see the answer regarding real world validation for `9j5r`.
>
> ### Exploration of scalability to larger production systems
>
> In general, the scope of our paper was to introduce LineFlow and show that optimal policies can be
> learned in controllable scenarios. The scalability to larger production systems requires a more sophisticated application of RL algorithms than currently done in our work, and we want to leave this open for future research.
>
> Reviewer `9j5R` also asked about larger production systems and the curse of dimensionality, and
> we would like to refer to our answer *Multi-agent RL* given to `9j5R`. We hope that also
> provides clarification for this point raised.
>
> ### Heterogeneous machine types
>
> Yes, LineFlow can handle heterogeneous machine types, and our experiments reflect this (see also
> Appendix B). We included scenarios where machines have varying processing times or assembly
> processes—key characteristics of heterogeneous machine types.
>
> ### Goodness-of-fit for ICML
>
> We appreciate the reviewer’s acknowledgment of the solid engineering effort behind LineFlow and the
> bridging of reinforcement learning with production line optimization. At the same time, we take the
> concern regarding the lack of algorithmic innovation seriously and are grateful for the honest and
> constructive feedback. We also want to apologize if the use of the phrase *theoretical foundation* may
> have given the impression that the paper aims to introduce novel theoretical results. Our intention
> was to refer to the theoretical validation of the benchmark tasks—by computing ground-truth
> optima—rather than to suggest the development of new RL theory. We recognize that this wording may
> have caused confusion and will revise the manuscript to clarify our intent and better reflect the
> nature of our contributions.
>
> We acknowledge that our work primarily focuses on applying reinforcement learning to an industrial
> problem rather than introducing a completely new RL algorithm. However, we believe that our
> contributions align well with ICML’s scope as the methodological contributions and real-world impact
> of our work are valuable to the ICML community. Many impactful papers at ICML focus on novel
> applications, domain-specific adaptations, or applying RL to new challenges, and we believe our work
> fits within this tradition.

---

### Official Review · Reviewer_Hikp · 2025-03-13

**Overall Recommendation:** 1

**Summary:**

The paper introduces LineFlow, a reinforcement learning-based framework for optimizing production line reallocation, rescheduling, and routing. The authors develop LineFlow as an extensible Python package that facilitates large-scale RL training and simulation. They benchmark multiple RL algorithms, including PPO, TRPO, and A2C, demonstrating that RL-based policies can approximate optimal solutions in structured scenarios. The study presents three key case studies—optimal waiting time, part distribution, and worker assignment—alongside a more complex case that integrates these factors. The work is application-driven, offering insights for both manufacturing and RL-based decision-making.

**Claims And Evidence:**

The submission provides empirical support for its claims, but key justifications are lacking. Given the queuing nature of production lines, the choice of a discrete-time MDP over a continuous-time MDP is not adequately justified. Similarly, the reward function lacks a clear rationale. These gaps raise doubts about the practical justifiability of the results.

**Essential References Not Discussed:**

N/A

**Experimental Designs Or Analyses:**

1. It is unclear why the problem is modeled as a discrete-time MDP rather than a continuous-time MDP, given the queuing dynamics of production lines.
2. The paper does not justify the chosen reward function, leaving it uncertain whether it accurately reflects real-world optimization objectives.  The rationale for framing the control objective (maximizing Cost while minimizing time to reach the maximum) within a discounted optimization framework needs further explanation.
3. It seems that paper uses another approach to compute optimal layout. The complexity of that algorithm, and its performance against RL is not provided.

**Methods And Evaluation Criteria:**

Same as above.

**Other Comments Or Suggestions:**

N/A

**Other Strengths And Weaknesses:**

Strengths:

- The paper introduces a Python package for simulating production lines and training RL agents.
- It evaluates multiple RL algorithms (PPO, TRPO, A2C) across diverse production scenarios.

Weaknesses:

- It lacks justification for using a discrete-time MDP over a continuous-time MDP.
- The formulation of rewards and the optimization objective (discounted vs. average vs. total reward) lacks motivation, raising doubts about real-world alignment.
- The paper focuses on practical applications without offering broader theoretical insights for similar problems or presenting novel challenges for the broader ML community.
- It does not convincingly argue for the necessity of RL over traditional optimization methods.

**Questions For Authors:**

1. Given this production line setting, can probability distributions over time delays be ignored? Specifically, why is the system modeled as a discrete-time MDP?
2. If the goal is to "find a control policy π that maximizes Cπ(t) while simultaneously minimizing the time t to reach the maximum," how is this reduced to bounded-time reward maximization? What is the exact optimization problem being solved? Is it an average-reward (throughput maximization) problem, or a reachability-like problem where the objective is to achieve the maximum value as quickly as possible?
3. What algorithm is used to compute the optimal reward? What is its complexity, and how does it compare to RL?

**Relation To Broader Scientific Literature:**

N/A

**Theoretical Claims:**

N/A

---

> ### Author Rebuttal · Authors · 2025-03-31
>
> We would like to thank the reviewer for the insightful feedback and
> suggestions.
>
> ### Justification of discrete-time MDP
>
> A major concern of the reviewer was why we used a discrete-time MDP (DTMDP) to model active line control
> problems instead of a continuous-time MDP (CTMDP), and we thank the reviewer for this insightful and
> important comment.
> We want to highlight that in LineFlow, the simulation runs in continuous time and only the
> interaction with the agents is discretized.
> We acknowledge that production lines typically operate in continuous time, and a CTMDP formulation
> could, in principle, more naturally capture the queuing dynamics. However, we opted for a
> discrete-time MDP for the following reasons:
>
> - **Control Intervals in Production Systems**: In real-world manufacturing, decision-making often
>   happens at fixed intervals (e.g., every few seconds or minutes), corresponding to worker task
>   assignments, batch processing schedules, or supervisory interventions. Our DTMDP formulation
>   aligns with these real-world control policies.
> - **Discretization of Continuous Events**: While production events (e.g., job completions) occur
>   asynchronously, they can be approximated using a sufficiently fine-grained discrete-time model
>   without significant loss of fidelity. Similar approaches have been used in prior RL research for
>   industrial systems. This is accomplished by the parameter `step_size` in the LineFlow
>   implementation.
> - **Practicality in RL Training**: Many reinforcement learning algorithms, particularly deep RL
>   methods like PPO, are naturally formulated in discrete time.
>   Extending them to continuous-time settings requires additional modifications with unclear practical benefits and more complexity on the agent's side.
>
> We appreciate this perspective and will expand our discussion in Section 3.1 of the revised
> manuscript to clarify the trade-offs between discrete-time and continuous-time modeling. Moreover,
> we also would like to point the reviewer to our reply to `9j5R` about *Real-World validation*, where
> one also can see how well the discrete MDP approximate a real-world system.
>
> ### RL vs traditional optimization methods
>
> The reviewer wrote:
>
> > It seems that the paper uses another approach to compute optimal layout. The complexity of that algorithm, and its performance against RL is not provided.
>
> and
>
> > It does not convincingly argue for the necessity of RL over traditional optimization methods.
>
> It seems there may be a misunderstanding at the reviewers side. If by *optimal layout* the reviewer means *optimal policy*, the optimal policies for the case studies are derived by solving the mathematical optimization problems in Section 4, as detailed in Section C.
>
> If by *optimal layout* the reviewer refers to layout optimization, then we have to emphasize that our work is
> **not** focused on this. For this topic, established methods like assembly line balancing
> already exist. Instead, we aim to find a policy $\pi$ maximizing
> $\mathbb{E}[C_{\pi}(T_{\mathrm{sim}})]$ under the MDP dynamics.
> We are not aware of any standard approach adressing this task, and we welcome pointers to
> relevant literature (please, also see our reply to reviewer `R3X2` on the **Comparison with
> industry-standard approaches**).
>
> ### Justification of reward function
>
> The reviewer wrote:
>
> > The formulation of rewards and the optimization objective (discounted vs. average vs. total reward) lacks motivation, raising doubts about real-world alignment.
>
> We thank the reviewer for the comment and apologize for any lack of clarity. In
> Section 2.3, we elaborate how our reward structure is based on Overall Equipment
> Effectiveness (OEE), a **widely-used** manufacturing metric, ensuring alignment
> with industry objectives and providing an interpretable signal for policy
> learning. We further chose to optimize total reward over a fixed time horizon (e.g., an
> 8-hour shift) to reflect typical planning cycles, distributing it temporally to
> facilitate learning.
>
> We acknowledge that a more explicit discussion of these design decisions could help
> strengthen the paper and will revise Section 2.3 accordingly.

---

### Official Review · Reviewer_A1PN · 2025-03-21

**Overall Recommendation:** 3

**Summary:**

This paper proposes LineFlow, an environment construction framework for production lines, which provides a generalized framework for research in the field of production lines. Additionally, it constructs several typical and complex scenarios to evaluate the performance of different reinforcement learning algorithms.

**Claims And Evidence:**

Yes.

**Essential References Not Discussed:**

Some recent related works concerning the production line problem have been involved.

**Experimental Designs Or Analyses:**

Yes. I checked the experiments setting and the results, and did not find main flaws.

**Methods And Evaluation Criteria:**

Yes. This work gives the production line design in detail.

**Other Comments Or Suggestions:**

**Suggestions:**

The main text includes many specific layout diagrams, while the introduction to the LineFlow framework itself is minimal. It might be better to include the component diagram of LineFlow (Figure 9 in Appendix A.5) in the main text instead.



**Grammars:**

Check the grammar: "We refer to Section A.1 for more details about the worker assignment is implemented in LineFlow."

**Other Strengths And Weaknesses:**

**Strengths:**

Regardless of the domain knowledge related to production lines, this paper is relatively easy to follow. It thoroughly considers several typical cases in active line control, introduces fundamental elements, constructs the LineFlow framework, and encapsulates it in the Gym interface, which is convenient for RL researchers. The LineFlow framework allows for easy customization of new scenarios and offers flexible interfaces.



**Weakness:**

- Some statements are slightly unclear (see questions).
- The authors only consider the *processing time* of stations and their *statistical interplay*, assuming this time follows an exponential distribution. This design may oversimplify real-world scenarios.
- The action space design is only briefly discussed. Training an RL agent in a large action space is challenging. If the agent must distribute parts or workers to different stations at scale, this will lead to a complex combinatorial optimization problem and face the notorious curse of dimensionality. Consequently, LineFlow’s running efficiency may be hindered. I also noted that the training time for a single model is non-negligible (approximately 14 hours for PD$_5$ and 6 days with an H100 GPU in CL for recurrent PPO). Thus, how to reduce the action space and how LineFlow addresses this issue should be clearly explained.

**Questions For Authors:**

1. Do OK parts mean parts that have been finished and NOK mean not OK?
2. What are the action spaces, and how to ensure the actions are feasible? In production line problems, actions should satisfy certain constraints. For example, in Sec. 4.3 Eq.(2), worker assignment is inherently an integer partition optimization problem.
3. How is the running time of each step in the three cases and in CL?
4. Is there any possibility of other distributions for the processing time?
5. Why can the best rewards in WT, WTJ, WA and PD match the optimal rewards while falling far behind the "simple" heuristic baselines in CL?

**Relation To Broader Scientific Literature:**

This work may contribute to RL for combinatorial optimization, especially for scheduling  or VRP, which are classical NP-hard problems.

**Theoretical Claims:**

N/A. This paper does not include theoretical claims. The proofs are provided for the optimality of the computation in 3 cases.

---

> ### Author Rebuttal · Authors · 2025-03-31
>
> We thank the reviewer for the very helpful comments and suggestions that helped us improve our
> manuscript in various places.
>
> ### Replies to strengths and weaknesses
>
> > The authors only consider the processing time of stations and their statistical interplay, assuming this time follows an exponential distribution. This design may oversimplify real-world scenarios.
>
> Before replying, we want to emphasize that both restrictions are not limitations of LineFlow in
> general, but of the specific experiments we performed. In fact, LineFlow allows users to set an individual
> and specific statistical distribution for each process. Details will be released in the documentation of LineFlow.
> The fact that we used the exponential distribution to model the processing time is because this is
> common in many statistical analyses of production lines, such as in [2]. We validated our assumption with
> the dataset [1] and found that the exponential distribution constitutes a reasonable assumption here
> (see also our reply to reviewer `9j5R` for more details on the analysis).
>
> Regarding the action space: The actions possible depend on the objects used in the layout and, as correctly identified by the
> reviewer, also scale correspondingly. However, we see that for production lines, the action space
> is not directly something that can be explicitly *designed* but is more a consequence of the
> concrete layout, which we considered as fixed. We apologize that this was not addressed well enough
> in the initial version of our manuscript, and we will expand Section 3.2 in the revised version.
>
> Consequently, we are of the opinion that the curse of dimensionality is less a problem of LineFlow
> in particular, but more an inherent challenge of *RL for active line control*. As correctly
> identified by the reviewer, this is also what we see in our experiments: Finding optimal control
> policies for larger layouts requires significantly more training time. We think this is a key challenge
> for further RL research that can be addressed using LineFlow.
>
> ### Replies to specific questions
>
> > Do OK parts mean parts that have been finished and NOK mean not OK?
>
> We apologize that we have not elaborated on these abbreviations in more detail, and we will provide a
> precise definition in the revised manuscript. In general, OK denotes parts successfully built into a
> final product, while NOK parts denote parts that have been discarded at some station of the line.
>
> Regarding to keep the actions feasible: In fact, one key benefit of LineFlow is that the action
> space is a consequence of the layout to be optimized, and it is impossible for a given policy to
> violate constraints induced on the action space. This is due to the fact that we designed LineFlow
> in a way to model elementary objects of assembly lines to match their real-world behavior. In the
> concrete case of WA, every workers is modeled as a specific objects that can be assigned
> independently to at most stations. We agree with the reviewer that the action space of the worker
> assignment problem could be presented in more detail, and we will improve this in the revised
> version.
>
> > How is the running time of each step in the three cases and in CL?
>
> The times per step are (k=3, 4, 5):
> - WT/WTJ: 0.000052s/ 0.000044s
> - PD: 0.00011s, 0.00013s, 0.00016s
> - WA: 0.00016, 0.00018s, 0.0002s
> - CL: 0.00008s, 0.00010s, 0.00012s
>
> > Why can the best rewards in WT, WTJ, WA, and PD match the optimal rewards while falling far behind the "simple" heuristic baselines in CL?
>
> We think the key reason is that the CL scenario combines multiple control challenges — worker
> assignment, component distribution, waiting time tuning, and timing-sensitive scrap avoidance — into a
> single high-dimensional task that requires long-term memory and coordinated action sequences. While
> the *atomic* tasks each isolate one of these dimensions, CL requires simultaneous control across all
> of them, making it significantly harder to learn from scratch.
>
> In contrast, the heuristic baseline for CL incorporates prior domain knowledge and is hand-tuned to
> balance part flow and buffer states in a globally consistent way. This prior knowledge allows the
> heuristic to avoid deadlocks and high scrap rates early on, while RL agents typically suffer from
> poor initial exploration, leading to unproductive or even blocking behaviors during early training
> (as shown in Figure 6).
>
> We will revise the discussion in Section 5.3 to clarify this gap and to better contextualize the
> strengths and current limitations of the RL approach in complex tasks.
>
> ### References:
> - [1] Meg Risdal et. al., Bosch Production Line Performance. https://kaggle.com/competitions/bosch-production-line-performance, 2016.
>   Kaggle.
>
> - [2] Bierbooms, R. (2012). *Performance Analysis of Production Lines: Discrete and Continuous Flow Models*. PhD Thesis, Technische Universiteit Eindhoven, Eindhoven. ISBN: 9789053355435.

---

> > ### Comment · Reviewer_A1PN · 2025-04-08
> >
> > Thanks for the clarification.

---

> > > ### Author Response · Authors · 2025-04-08
> > >
> > > We are glad that our clarification helps. Given that we addressed your primary concerns raised in the review, we would kindly ask you to reconsider your review score while taking the rebuttal into account.

---

### Decision · Program_Chairs · 2025-05-01

**Decision:**

Accept (poster)

**Comment:**

The paper has received an overall mixed set of reviews, with two positive recommendations and two more critical assessments. After carefully considering the reviews and the authors' rebuttal, I believe most of the major concerns have been adequately addressed.

Reviewer Hikp raised questions regarding some unclear elements, which seem to have been largely addressed.
Reviewer R3X2's main critique was that the paper "applies standard optimization principles to specific production settings", which is a fair point when considering the novelty threshold for publication. However, the reviewer also acknowledged several strengths in the work (well-written, comprehensive framework, appropriate methodology).

In summary, the quality of the paper is generally appreciated by the reviewers and while not highly novel, this paper brings a useful framework for research in the field. I recommend a weak accept.